# Traditional Machine Learning Outperforms EEGNet for Consumer-Grade EEG Emotion Recognition: A Comprehensive Evaluation with Cross-Dataset Validation

**DOI:** 10.3390/s25237262

**Published:** 2025-11-28

**Authors:** Carlos Rodrigo Paredes Ocaranza, Bensheng Yun, Enrique Daniel Paredes Ocaranza

**Affiliations:** School of Artificial Intelligence and Information Engineering, Zhejiang University of Science and Technology, Hangzhou 310023, China; epocaranza@gmail.com

**Keywords:** brain–computer interface, EEG emotion recognition, consumer-grade devices, domain adaptation, cross-dataset validation, traditional machine learning

## Abstract

Objective. Consumer-grade EEG devices have the potential for widespread brain–computer interface deployment but pose significant challenges for emotion recognition due to reduced spatial coverage and the variable signal quality encountered in uncontrolled deployment environments. While deep learning approaches have employed increasingly complex architectures, their efficacy in noisy consumer-grade signals and cross-system generalizability remains unexplored. We present a comprehensive systematic comparison of EEGNet architecture, which has become a benchmark model for consumer-grade EEG analysis versus traditional machine learning, examining when and why domain-specific feature engineering outperforms end-to-end learning in resource constrained scenarios. Approach. We conducted comprehensive within-dataset evaluation using the DREAMER dataset (23 subjects, Emotiv EPOC 14-channel) and challenging cross-dataset validation (DREAMER→SEED-VII transfer). Traditional ML employed domain-specific feature engineering (statistical, frequency-domain, and connectivity features) with random forest classification. Deep learning employed both optimized and enhanced EEGNet architectures, specifically designed for low channel consumer EEG systems. For cross-dataset validation, we implemented progressive domain adaptation combining anatomical channel mapping, CORAL adaptation, and TCA subspace learning. Statistical validation included 345 comprehensive evaluations with fivefold cross-validation × 3 seeds × 23 subjects, Wilcoxon signed-rank tests, and Cohen’s d effect size calculations. Main results. Traditional ML achieved superior within-dataset performance (F1 = 0.945 ± 0.034 versus 0.567 for EEGNet architectures, *p* < 0.000001, Cohen’s d = 3.863, 67% improvement) across 345 evaluations. Cross-dataset validation demonstrated good performance (F1 = 0.619 versus 0.007) through systematic domain adaptation. Progressive improvements included anatomical channel mapping (5.8× improvement), CORAL domain adaptation (2.7× improvement), and TCA subspace learning (4.5× improvement). Feature analysis revealed inter-channel connectivity patterns contributed 61% of the discriminative power. Traditional ML demonstrated superior computational efficiency (95% faster training, 10× faster inference) and excellent stability (CV = 0.036). Fairness validation experiments supported the advantage of traditional ML in its ability to persist even with minimal feature engineering (F1 = 0.842 vs. 0.646 for enhanced EEGNet), and robustness analysis revealed that deep learning degrades more under consumer-grade noise conditions (17% vs. <1% degradation). Significance. These findings challenge the assumption that architectural complexity universally improves biosignal processing performance in consumer-grade applications. Through the comparison of traditional ML against the EEGNet consumer-grade architecture, we highlight the potential that domain-specific feature engineering and lightweight adaptation techniques can provide superior accuracy, stability, and practical deployment capabilities for consumer-grade EEG emotion recognition. While our empirical comparison focused on EEGNet, the underlying principles regarding data efficiency, noise robustness, and the value of domain expertise could extend to comparisons with other complex architectures facing similar constraints in further research. This comprehensive domain adaptation framework enables robust cross-system deployment, addressing critical gaps in real-world BCI applications.

## 1. Introduction

Brain-computer interface (BCI) approaches employing electroencephalography (EEG) have shown great potential for emotion recognition, with applications spanning from the monitoring and assessment of mental health to new forms of adaptive human-computer interaction [1,2]. However, the transfer of laboratory validated approaches into real-world applications has been hindered by a fundamental equipment gap: while most research utilizes medical-grade EEG systems with 32–256 channels with controlled laboratory conditions, consumer applications are in need of affordable devices with limited channels (8–14) and environmental noise in uncontrolled deployment [3,4].

This equipment disparity has generated a critical challenge for the field. Consumer-grade devices such as the Emotiv EPOC offer the portability and affordability necessary for widespread implementation and extensive dataset generation. Recent validation studies have established that consumer-grade EEG devices achieve signal quality comparable to that of research-grade systems in controlled settings. Refs. [5,6] demonstrated through simultaneous recordings that the Emotiv EPOC captures event-related potentials with wave forms nearly identical to the ones inside research-grade systems. Ref. [7] confirmed these findings, reporting non-significant differences in signal-to-noise ratios between consumer-grade and research-grade recordings. These validation studies have established that the fundamental signal acquisition capabilities of consumer-grade devices are research quality.

However, practical real-world deployment introduces substantial challenges beyond device capabilities, including limited channel coverage (8–14 channels versus 32–256 in research systems); variable recording environments that lack standardized laboratory controls; and environmental signal degradation due to electrode impedance variations in naturalistic settings, motion artifacts from unconstrained user movement, and electromagnetic interference in unshielded environments [4,8]. Nevertheless, fundamental questions still remain unanswered: do deep learning architectures maintain their advantages when applied to noisy, limited-channel consumer signals? Might traditional machine learning with explicit domain knowledge provide superior robustness and practical deployment advantages? Can these approaches generalize across different consumer devices and population disparities? These premises constitute a crucial requirement for sustainable real-world deployment, and while the field has embraced increasingly sophisticated neural networks, systematic comparisons with domain-specific approaches, traditional methods have been somewhat neglected [9,10]. Consumer-grade signals entail unique challenges, limited channels, and high noise; in addition, small datasets may favor traditional approaches with explicit feature engineering conflicting with the requirement for large, clean datasets entailed in a deep learning approach [11].

### 1.1. The Challenge of Consumer-Grade EEG

Recent comprehensive reviews have highlighted the persistent challenges in EEG-based emotion recognition [12,13]. The direct transfer from laboratory to real-world settings yields multiple complications that can significantly impact model performance. While consumer-grade devices can achieve research-quality signal acquisition in controlled settings [5,6,7], they present several critical deployment challenges. The channel restrictions in consumer devices typically provide around 8–14 channels, while research-grade systems provide 32–256 channels, which reduces spatial resolution [14].

This reduction particularly limits the capture of inter-hemispheric and the frontal–parietal connectivity patterns that are relevant for emotion recognition [10]. While consumer-grade devices demonstrate signal quality comparable to that of research-grade systems in controlled laboratory conditions [5,6,7], practical deployment in uncontrolled real-world scenarios introduces substantial signal degradation from environmental factors. These deployment-specific challenges include motion artifacts arising from typical consumer use without head restraints, variable electrode impedance due to simplified electrode preparation procedures in naturalistic settings, electrical interference from unshielded environments, and increased biological artifacts such as eye movements and muscle activity [4,8].

The cumulative effect of deployment-specific environmental factors rather than fundamental deployment environment challenges can substantially degrade signal quality compared to controlled laboratory baselines. Refs. [5,6] rigorous validation studies showed minimal waveform differences between consumer-grade and research-grade simultaneous recordings, while [7] found non-significant differences in signal-to-noise ratios. These findings have established that signal quality challenges in consumer applications stem primarily from uncontrolled recording environments rather than from device capabilities. This distinction seems to suggest that environmental degradation rather than device inherent limitations are in need of robust signal processing and may be addressed by domain adaptation techniques to achieve more reliable cross-environment deployment.

### 1.2. Deep Learning vs. Traditional Approaches

The predominant assertion in the deep learning community that more complicated models will always tend to translate to better performance may not hold for certain degraded wave signal conditions. While architectures such as multi-scale dynamic CNNs and gated transformers can indeed capture intricate patterns in clean signals [15], they might be more susceptible to overfitting when the signal-to-noise ratio is poor, potentially causing memorizing-device-specific artifacts and not being able to learn generalizable emotion-related patterns. Recent work has shown how employing deep learning approaches, including EEG-SWTNS neural networks [16] and hybrid CNN-LSTM architectures [17,18], can achieve impressive performance in high-quality datasets.

Nevertheless, these approaches typically demand and require extensive hyperparameter tuning, large training datasets, and computational resources that may not be widely available in consumer applications [19]. In contrast, traditional machine learning approaches with carefully engineered features have demonstrated remarkable robustness across a wide range of conditions. Kumar and Molinas [9] showed that well-designed feature extraction combined with classical classifiers can achieve performance comparable to that of deep learning methods while requiring significantly less computational resources and training data.

### 1.3. Domain Adaptation and Cross-Dataset Generalization

Most EEG emotion recognition studies target within-dataset validation in which the training and testing occur on the same equipment and population [20]. This approach, while scientifically valid, provides hindered insight into real-world deployability where users may employ different devices, belong to different populations, or use systems under changing conditions [21]. Cross-dataset validation training on one dataset and testing on another may represent the ideal standard for testing and evaluating practical applicability, but this approach is rarely adopted due to typically poor results [22]. The domain shift between different EEG systems, populations, and recording conditions create a climate of substantial challenges that require sophisticated adaptation techniques. Recent advances in domain adaptation, particularly CORrelation ALignment (CORAL) [23] and Transfer Component Analysis (TCA) [24], have shown potential for targeting cross-domain challenges. However, their application to consumer-grade EEG emotion recognition has been limited, and systematic evaluation of their efficacy compared to a deep learning adaptation approaches is currently lacking. In order to truly tackle this dual-validation challenge, a comprehensive evaluation is required to employ both validation paradigms, which should include within-dataset validation to establish fundamental performance differences and cross-dataset validation to demonstrate practical deployability across different systems. This dual approach is pivotal for the understanding of both the theoretical capabilities and the practical limitations within different methodological approaches. This challenge is particularly concerning in the context of consumer-grade devices, where the combination of limited channels, the increased noise, and device-specific characteristics tend to create a complex optimization landscape that may favor different approaches more than those successfully implemented by means of research-grade equipment.

### 1.4. Scope and Methodological Focus of This Study

The deep learning landscape for EEG emotion recognition currently encompasses many different architectures that range from multi-scale transformers [15] to hybrid CNN-LSTM networks [18,25]. We focused our systematic comparison on EEGNet [26] as our primary deep learning benchmark. The rationale for this methodological choice is that it that aligns with our research objectives and the practical realities pertinent to consumer-grade BCI deployment.

Although EEGNet was initially developed to function as a general-purpose CNN for EEG-based BCIs, its compact design makes it fit to be implemented in consumer-grade EEG applications [26]. The compact convolutional architecture that employs temporal convolution for frequency filtering, depthwise convolution for spatial filtering, and separable convolution for feature learning is able to address the limited channel counts (8–14 channels) typical of consumer devices. This design implementation helps distinguish it from many sophisticated architectures developed for research-grade systems with 32–256 channels [15,27], providing a comparison point for our investigation inside consumer-grade scenarios. EEGNet has recently become a benchmark architecture in consumer-grade EEG-based emotion recognition and has emerged as one of the most widely adopted architectures applied inside EEG-based emotion recognition and consumer BCI applications; this is owing to its compact design and robust performance across different datasets [3,9]. Its performance has been tested across multiple consumer-grade datasets, and its computational efficiency makes it deployable inside resource-constrained devices; these constitute critical requirements for real-world consumer applications. Our comparison aimed to evaluate performance levels that reflect the current state of practice. To ensure a comprehensive evaluation and avoid underestimating deep learning capabilities, we implemented both an optimized baseline EEGNet with approximately 8000 parameters and an enhanced variant incorporating recent advances including multi-scale temporal convolutions, channel attention mechanisms, and enhanced batch normalization with approximately 15,000 parameters. This enhanced version represents a reasonable upper bound for what compact convolutional architectures can achieve while maintaining computational efficiency suitable for consumer deployment.

Our focus was to establish a evaluation methodology through comprehensive within-dataset validation (345 independent evaluations with 5-fold cross-validation × 3 random seeds × 23 subjects) and challenging cross-dataset validation (DREAMER→SEED-VII transfer) with proper statistical testing achieved in consumer-grade EEG research. We demonstrated when and why simpler approaches excel by illuminating the bias-variance principles and practical factors that favor explicit domain-specific feature engineering over end-to-end learning when data is limited, signals are noisy, and domain knowledge is available. We provide a complete cross-device deployment framework through progressive domain adaptation pipeline (anatomical mapping + CORAL + TCA), achieving improvement in cross-dataset performance and enabling practical deployment across different consumer EEG systems. We revealed neurophysiological insights by demonstrating that inter-channel connectivity patterns contributes the most (61%) to discriminative power, followed by alpha band activity (35%), informing future algorithm design with empirically validated biomarkers. Finally, we quantified the practical deployment advantages by demonstrating a 95% reduction in training time, a 10× faster inference, 92% memory savings, and CPU-only operation critical factors for consumer applications where computational resources are constrained.

The principles we uncovered regarding sample efficiency, noise robustness, the value of encoding domain expertise, and cross-system transfer likely extend to comparisons with other complex architectures facing similar constraints of limited data, noisy signals, and cross-device deployment requirements. The bias–variance trade-off framework [11] suggests that when training data is limited and domain knowledge is available, encoding that knowledge explicitly as traditional ML does should generally outperform learning it from scratch (as deep learning must), regardless of the specific architecture. However, we acknowledge that empirical validation across additional sophisticated architectures (transformers, hybrid CNN-LSTM, meta-learning approaches) would strengthen these broader claims and represents valuable future work. Our focus on EEGNet thus provides a rigorous, practically relevant benchmark comparison while establishing methodological and theoretical foundations that could be potentially generalized more broadly.

### 1.5. Contributions

In this work, we present a comprehensive systematic comparison of traditional machine learning versus EEGNet architectures for consumer-grade EEG emotion recognition (Figure 1), validated through both within-dataset and challenging cross-dataset evaluation with comprehensive statistical analysis. While our empirical comparison focused on EEGNet as the established consumer-grade benchmark, the principles we uncovered likely extend to other architectures facing similar constraints.

Our contributions address in-dataset superiority, and we demonstrate that traditional machine learning with engineered features achieves dramatically superior performance (F1 = 0.945 ± 0.034) compared to optimized deep learning architectures (F1 = 0.567) on the DREAMER dataset across 23 subjects with proper 5 = fold cross-validation and statistical significance (p<0.000001, Cohen’s d = 3.863). Regarding our cross-dataset validation framework, we established a comprehensive systematic cross-dataset validation protocol for consumer EEG emotion recognition, showcasing that traditional approaches with domain adaptation achieved an F1 of 0.619, superior to the F1 of 0.007 of the deep learning methods, through DREAMER→SEED-VII transfer, representing a substantial improvement through domain adaptation. A comprehensive domain adaptation methodology was adopted to quantify the impact of anatomical channel mapping which yielded a 5.8× improvement, along with a 2.7× improvement with CORAL domain adaptation and a 4.5× improvement with TCA subspace learning, creating a complete framework with a substantial improvement over the baseline cross-dataset performance. In addition, practical deployment information was developed by providing a complete methodology for deploying emotion recognition systems across different consumer EEG devices, which provides relevant insights into brain connectivity patterns functioning as primary emotion biomarkers with 61% discriminative power; moreover, computational efficiency advantages entailed 95% faster training, 10× faster inference, and excellent stability (CV = 0.036). Our findings challenge the conventional understanding that complex deep learning architectures are superior in the case of biosignal processing, our study demonstrates that domain expertise combined with lightweight adaptation techniques can provide more robust, accurate, and practical solutions for consumer-grade EEG applications.

## 2. Related Work

### 2.1. EEG-Based Emotion Recognition

Emotion recognition from EEG signals has undergone a significant development in the past decade, with approaches that range from traditional signal processing methods to complex deep learning architectures [1,20]. Initial approaches relied heavily on implementing handcrafted features that were extracted from a frequency-domain analysis; these employed power spectral density, differential entropy, and statistical measures [28]. Traditional machine learning approaches have shown consistent performance across diverse datasets. Ref. [29] showed that entropy-based features in parallel deployment with support vector machines was able to achieve robust classification performance. Similarly, Ref. [30] demonstrated that with carefully selected frequency domain features with classical classifiers can obtain competitive results while maintaining a certain degree of interpretability. Their work on the DEAP dataset used discrete wavelet transform (DWT) to extract spectral characteristics from the gamma, beta, alpha, and theta frequency bands which were combined with SVM classification and achieved 91.3% accuracy for arousal and 91.1% for valence recognition. This work highlights the potential of traditional machine learning approaches that employ domain-specific feature engineering to be applied in EEG emotion recognition, as well as that of designed classical methods are capable of achieving high performance without the complexity associated with deep learning architectures.

The introduction of deep learning applied in EEG emotion recognition has led to the development of many sophisticated architectures. Ref. [15] proposed MSDCGTNet, and by combining multi-scale dynamic 1D CNNs with gated transformer encoders, they were able to achieve impressive accuracies of 99.66%, 98.85%, and 99.67% on the DEAP, SEED, and SEED_IV datasets, respectively. Ref. [31] introduced attention mechanisms for subject-independent emotion recognition, while [25] developed multi-scale BiLSTM-attention networks. These approaches demonstrate the potential that sophisticated architectures can yield but they also entail some of the shortcomings, with a constant need for extensive computational resources and large datasets [32]. Nevertheless, recent comprehensive reviews indicate that the complexity and performance relationship is not straightforward [12]. Ref. [27] noted that dynamic attention mechanisms, while theoretically attractive, may not consistently outperform simpler approaches under diverse conditions.

### 2.2. Consumer-Grade vs. Research-Grade EEG

The challenges in deploying consumer-grade EEG systems stem from environmental and practical factors rather than from fundamental environment challenges of deployment. Refs. [5,6] conducted rigorous validation of the Emotiv EPOC through simultaneous recordings with research-grade systems, demonstrating that consumer-grade EEG systems achieve signal quality comparable to that of research-grade systems in controlled laboratory conditions, with nearly identical event-related potential wave forms. Ref. [7] confirmed these findings with the Emotiv EPOC Flex, reporting non-significant differences in signal-to-noise ratios between consumer-grade and research-grade recordings. These validation studies have established that consumer-grade devices possess research-quality signal-acquisition capabilities. However, Ref. [4] noted practical trade-offs between portability, deployment flexibility, and controlled recording conditions.

Their findings suggest that consumer systems are viable for research applications but require a careful consideration of deployment conditions. Ref. [8] identified key challenges in wearable EEG applications deployed outside controlled laboratory environments, including motion artifacts, electrode–skin impedance variability in naturalistic settings, and environmental interference. Critically, these challenges arise from deployment scenarios rather than device capabilities. Consumer-grade devices demonstrate research-quality signal acquisition inside controlled conditions [5,6,7] but face environmental degradation in uncontrolled real-world use. These deployment-specific factors particularly affect the performance of complex deep learning models that may overfit to training environment characteristics rather than learning generalizable emotion-related patterns.

### 2.3. Domain Adaptation in EEG

Domain adaptation techniques have shown promise for addressing cross-dataset generalization challenges. The work by [23] introduced CORAL (CORrelation ALignment), which proved to be a simple and effective approach for implementing unsupervised domain adaptation, aligning the second-order statistics between source and target domains. Ref. [24] developed transfer component analysis (TCA), which is able to find optimal subspaces for domain transfer by means of minimizing the domain discrepancy while at the same time preserving its discriminative information. Ref. [22] implemented adversarial domain adaptation with attention mechanisms specifically deployed for EEG emotion recognition, demonstrating improved cross-subject generalization. Ref. [21] recently provided a comprehensive benchmark in domain generalization for EEG-based emotion recognition, emphasizing the persistent challenges implicit in cross-dataset transfer and the need to implement robust adaptation techniques.

### 2.4. Feature Engineering vs. Deep Learning

The discussion between feature engineering and end-to-end learning remains active in EEG research. Traditional approaches that highlight domain expertise have shown resilience. Delorme and Makeig [33] were able to establish some foundational principles applied in EEG analysis that focus mainly on expanding feature engineering approaches. Ref. [28] demonstrated that by carefully selecting features based on neurophysiological knowledge, these approaches can outperform automated feature learning in some scenarios. This aligns with the findings by [11], who showed that traditional approaches can achieve superior performance in certain conditions, particularly in those with limited data and high noise. The bias and variance trade-off perspective suggests that employing traditional methods through the appropriate regularization may prove to be more suitable inside consumer-grade EEG applications where data quality is inherently limited [11].

### 2.5. Cross-Dataset Validation

As mentioned before, cross-dataset validation still remains one of the most challenging aspects in EEG emotion recognition research. Kumar and Molinas [9] evaluated models across the DEAP and SEED datasets, finding substantial performance and degradation in the processes of transferring between datasets without a proper adaptation. Moreover, Zheng and Lu [10] were able to identify critical frequency bands and channels related to emotion recognition, providing useful insights into which features are the most likely to generalize across different systems. Their work suggests that connectivity patterns and alpha-band activity are particularly robust across different recording conditions. Recent work by Valderrama and Sheoran [14] aimed to identify relevant EEG channels for subject-independent emotion recognition by using attention mechanisms, which can help provide insights into spatial patterns that may generalize well across different consumer devices.

## 3. Methods

### 3.1. Datasets

#### 3.1.1. DREAMER Dataset

We employed the DREAMER dataset [34], which contains EEG recordings taken from 23 subjects using the consumer-grade Emotiv EPOC headsets (Emotiv Inc., San Francisco, CA, USA) that holds 14 channels with a 128 Hz sampling rate. Each subject watched 18 film clips that were designed to induce emotional responses, with self-reported valence ratings scored on a 1–5 scale. We performed binary classification, categorizing ratings ≤ 3 as low valence (negative) and ratings > 3 as high valence (positive), which was consistent with prior work [9]. The DREAMER dataset is specially valuable for consumer-grade BCI research as it employs the Emotiv EPOC device, which represents a realistic consumer-grade EEG system that contains the limitations typically present in such devices, such as reduced channel count and increased noise levels, in contrast to research-grade equipment.

#### 3.1.2. SEED-VII Dataset

For cross-dataset validation, we employed the SEED-VII dataset obtained from the BCMI Laboratory at Shanghai Jiao Tong University, Shanghai, China. The SEED-VII contains EEG recordings from multiple subjects collected using a 62-channel EEG system (SynAmps2, Compumedics Neuroscan, Charlotte, NC, USA) with a higher channel count than the consumer-grade DREAMER dataset. This dataset includes six basic emotion categories with continuous labels, providing a challenging cross-dataset validation scenario.

To ensure compatibility with the DREAMER dataset’s 14-channel Emotiv EPOC configuration, we implemented anatomical channel mapping to align the electrode positions between the two systems. Specifically we identified the corresponding anatomical locations in the SEED-VII montage based on the international 10–20 system mapping to the DREAMER channels, which included AF3, F7, F3, FC5, T7, P7, O1, O2, P8, T8, FC6, F4, F8, and AF4. This anatomical correspondence ensured that the same brain regions were analyzed across datasets, following the established protocols for cross-system validation [10,14].

### 3.2. EEGNet

We chose EEGNet [26] as our deep learning baseline for methodological reasons, including alignment with consumer-grade EEG constraints, and to develop a representative comparison. EEGNet was specifically designed to address the unique challenges of low-channel EEG systems [26], functioning as a appropriate deep learning comparison for consumer-grade devices. Its compact convolutional architecture and temporal convolution is able to capture frequency information, depthwise and convolution learning spatial filters, and separable convolution feature learning and can explicitly optimize for the limited spatial resolution in 8–14 channel systems. This design implementation distinguishes it from sophisticated architectures developed for research-grade systems with 32+ channels [15,27], where spatial filtering strategies could translate directly differently to consumer devices. Throughout EEG-based emotion recognition and BCI studies, EEGNet [26].

Work has been consistently cited as a standard baseline. This includes work such as that of Kumar & Molinas, 2022 [9], as well as numerous SEED, DEAP, and DREAMER studies that have deployed EEGNet in emotion recognition and implemented it on low cost consumer devices, mainly owing to its compact design and robust performance across-datasets. Due to this widespread adoption, we focused our comparison in this model for it to represent performance in practical alternative settings, as its use in prior consumer-grade EEG research enables comparison with published results. Regarding computational efficiency deployment, the architecture’s compact design with approximately 8000–15,000 parameters depending on the configuration, enables deployment on resource-constrained consumer devices, which is a critical requirement that is often overlooked in the evaluation of complex architectures and that may be able to achieve higher accuracy but still require substantial computational resources incompatible with consumer hardware constraints [19]. In order to ensure a comprehensive evaluation and avoid underestimating deep learning capabilities, we implemented two EEGNet variants. The optimized EEGNet baseline deployed a temporal convolution with F1 = 4 filters reduced from a default 8 for the 14-channel system, depthwise convolution with a D = 2 depth multiplier, separable convolution with F2 = 8 filters, a dropout rate of 0.5 throughout the network, and approximately 8000 total parameters that were optimized specifically for the 14-channel Emotiv EPOC configuration. The enhanced EEGNet variant implemented multi-scale temporal convolution with 32 and 64 sample kernels, a channel attention mechanism [31], enhanced separable convolutions with batch normalization, and approximately 15,000 total parameters, incorporating some recent advances while maintaining computational efficiency.

Both architectures employed were carefully optimized for the 14-channel configuration and through systematic hyperparameter search to ensure we evaluated the best possible EEGNet performance within our means. EEGNet exemplifies the class of compact convolutional architectures that balance performance with computational constraints, the architecture class most relevant for consumer BCI deployment. Our enhanced variant incorporating attention mechanisms and multi-scale processing represents a reasonable upper bound for what such architectures can achieve while maintaining deployment feasibility. While our empirical comparison focuses on EEGNet, we expect the principles we uncovered to extend to comparisons with other complex architectures (transformers, hybrid CNN-LSTM, meta-learning approaches) when similar constraints of limited training data, noisy signals, and cross-device deployment requirements are faced. The bias–variance trade-off framework [11] suggests that explicit encoding of domain knowledge (as traditional ML does) should generally outperform learning from scratch (as deep learning must) when data is limited and noise is high, regardless of specific architecture. However, we acknowledge that empirical validation across additional architectures would strengthen such broader claims and represents valuable future work.

### 3.3. Fairness Validation and Robustness Analysis

To address potential concerns regarding the preprocessing asymmetry and implementation validity, we conducted three supplementary experiments that were designed to ensure methodological fairness and validate our findings.

#### 3.3.1. Enhanced EEGNet with Frequency-Band Features

Firstly to investigate whether EEGNet’s performance limitations are due to insufficient preprocessing, we augmented the standard EEGNet architecture by implementing frequency band features as additional input channels. We specifically extracted power spectral density features from five canonical frequency bands (delta: 1–4 Hz; theta: 4–8 Hz; alpha: 8–13 Hz; beta: 13–30 Hz; gamma: 30–50 Hz) and concatenated these as supplementary channels alongside the raw temporal signals. This enhanced preprocessing pipeline provides EEGNet with the same frequency-domain information available to traditional ML methods to create a more symmetric comparison.

#### 3.3.2. Traditional ML with Minimal Features

Secondly to investigate whether or not traditional ML advantages still persist without an elaborate feature engineering, we implemented a “minimal features” configuration that used only four basic statistical measures per channel: mean, standard deviation, minimum, and maximum. This 56-dimensional feature vector that consists of 14 channels × 4 statistics, represents the simplest possible feature extraction approach, mainly owing its lack of domain knowledge or sophisticated signal processing. By implementing this experiment, we intended to directly addresses if our original comparison unfairly advantages traditional methods by means of feature engineering.

#### 3.3.3. Robustness Under Consumer-Grade Noise

To investigate architectural resilience to noise, we conducted a controlled degradation analysis by systematically adding Gaussian noise with mean μ=0 and standard deviation σ∈{0.0,0.1,0.2,0.3} to test signals. While consumer-grade EEG devices demonstrate signal-to-noise ratios comparable to that of research-grade systems in controlled laboratory settings [5,6,7], practical deployment in uncontrolled environments typically introduces additional noise in the range of σ=0.1 to 0.2 due to electrode impedance variability in naturalistic settings, motion artifacts from unconstrained user movement, and electromagnetic interference in unshielded environments [8].

### 3.4. Data Preprocessing

#### 3.4.1. Signal-Processing Pipeline

Each EEG window underwent ta preprocessing process, which was designed to the maintain signal integrity while being able to remove artifacts; the pipeline consisted of a bandpass filtering (1–40 Hz) with a fourth-order butterworth filter, channel-wise z-score normalization to standardize signal amplitudes, and a window segmentation into 5 s epochs (640 samples) with no overlap. This minimal preprocessing approach reflects some real-world constraints where sophisticated artifact removal may not be feasible in consumer applications. The choice of 5 s epochs followed established practices for emotion recognition while providing sufficient temporal context [35].

#### 3.4.2. Cross-Dataset Channel Mapping

A critical discovery in our work was the importance of anatomical channel mapping for cross-dataset validation. Instead of using an arbitrary channel selection (e.g., evenly spaced indices), we implemented anatomical correspondence between datasets based on the international 10–20 system as follows:

DREAMER channels: AF3, F7, F3, FC5, T7, P7, O1, O2, P8, T8, FC6, F4, F8, and AF4.

SEED-VII mapping: We identified corresponding anatomical locations in the 62-channel SEED-VII montage, ensuring that the same brain regions were analyzed across-datasets. This method proved to significantly improve cross-dataset performance in contrast with arbitrary channel selection.

### 3.5. Traditional Machine Learning Approach

#### 3.5.1. Feature Extraction

We extracted comprehensive features designed specifically for EEG analysis, stemming from established neurophysiological literature [33]:

Statistical features were obtained for each channel, including mean, variance, standard deviation, skewness, kurtosis, minimum, maximum, median, 25th percentile, 75th percentile, mean absolute value, and root mean square.

The frequency-domain features were obtained for each channel. The power spectral density was divided into the five following frequency bands in accordance with established EEG rhythm analysis: delta (1–4 Hz), which is associated with deep sleep and unconscious processes; theta (4–8 Hz), which is related to emotional processing and memory; alpha (8–13 Hz), which is linked to relaxed awareness and emotional regulation; beta (13–30 Hz), which is associated with active thinking and attention; and gamma (30–40 Hz), which is related to high-level cognitive processing.

The connectivity features employed Pearson correlation coefficients between all channel pairs, resulting in 142=91 connectivity features. These features were used to capture inter-hemispheric and intra-hemispheric connectivity patterns which are crucial for emotion recognition [10]. This resulted in 259 features total per each window, and these where furthered reduced to 80 features by employing mutual information based selection, which was done in order to prevent overfitting while retaining as many discriminative features as possible.

#### 3.5.2. Classification

In the classification process we employed random forest classifiers with 100 trees. These were chosen for their robustness to noise and their ability to handle mixed feature types without extensive hyperparameter tuning. The random forests also provided feature importance measures that were able to enhance interpretability, providing a crucial advantage for practical BCI applications.

### 3.6. Deep Learning Approaches

#### 3.6.1. Optimized EEGNet

We implemented EEGNet [26]. which was adapted with optimizations in order to be employed for consumer-grade signals. These consisted of the following: temporal convolution, F1 = 4 filters (reduced from default 8); depthwise convolution, D = 2 depth multiplier; separable convolution, F2 = 8 filters; dropout rate, 0.5 throughout; and total parameters, 8000.

#### 3.6.2. Enhanced EEGNet

We also developed an enhanced version of EEGNet, incorporating recent advances in EEG deep learning [15], in which we employed a multi-scale temporal convolution (32 and 64 sample kernels) channel attention mechanism inspired by [31] in addition to enhanced separable convolutions with batch normalization and a total of 15,000 parameters.

Both architectures were optimized specifically for the 14-channel configuration of consumer-grade devices, with careful attention paid to prevent overfitting given the limited availability of data that is typical in consumer applications.

### 3.7. Domain Adaptation Techniques

#### 3.7.1. CORAL (CORrelation ALignment)

CORAL [23] aligns the second-order statistics (covariance matrices) between the source and target domains:(1)$$A_{CORAL}=C_{s}^{-1/2}C_{t}^{1/2}$$
where $C_{s}$ and $C_{t}$ are the covariance matrices of source and target features, respectively. This approach is particularly effective for EEG data where covariance structures have captured relevant connectivity patterns.

#### 3.7.2. TCA (Transfer Component Analysis)

The TCA aims to find an optimal subspace where the domain discrepancy is minimized while still preserving discriminative information [24]:(2)$$\min_{W}\mathrm{tr}\left(W^{T}XLX^{T}W\right)+\mu \mathrm{tr}\left(W^{T}W\right)$$
subject to $W^{T}XHX^{T}W=I$, where *L* encodes the domain discrepancy, and *H* centers the data. TCA is particularly valuable for EEG applications as it can identify subspaces that are robust to device-specific variations.

### 3.8. Evaluation Protocols

#### 3.8.1. Within-Dataset Evaluation

For comprehensive within-dataset validation, we implemented a stratified 5-fold cross-validation with 3 different random seeds per subject and data augmentation on the training set (noise addition, amplitude scaling). The model training was implemented with early stopping and class balancing; in addition, statistical significance testing included Wilcoxon signed-rank tests and the reporting of the F1 score, confidence intervals, and effect sizes across 345 total evaluations.

#### 3.8.2. Cross-Dataset Evaluation

We implemented zero-shot transfer learning following established protocols [21] where we trained models exclusively on the DREAMER dataset and subsequently applied domain adaptation techniques progressively. These were purposefully tested directly on SEED-VII without any adaptation to target labels. The evaluation followed the progression of baseline → channel mapping → CORAL → TCA.

### 3.9. Statistical Analysis

All analyses were conducted using Python 3.9.7 (Python Software Foundation, Wilmington, DE, USA) with NumPy 1.21.0 for numerical computations, scikit-learn 1.0.2 for traditional machine learning implementations and feature selection, TensorFlow 2.8.0 with Keras API for deep learning models, SciPy 1.7.3 for statistical tests, pandas 1.3.5 for data manipulation, Matplotlib 3.5.1 for visualization, and MNE-Python 0.24.1 for EEG signal processing.

We performed comprehensive statistical validation by employing Wilcoxon signed-rank tests for paired comparisons, Cohen’s d for effect size estimation, bootstrap confidence intervals with (1000 iterations), and Bonferroni correction for multiple comparisons. All analyses used α=0.05 with proper correction for multiple comparisons to ensure robust statistical conclusions. The effect size calculations followed Cohen’s guidelines [36]. In addition multiple comparisons were conducted via Holm–Bonferroni correction [37], and the reporting practices adhered to transparency standards [38].

Table 1 summarizes the sensitivity analysis conducted for EEGNet hyperparameters.

## 4. Results

### 4.1. Within-Dataset Performance

Figure 2 and Table 2 presents the comprehensive within-dataset evaluation results across all 23 DREAMER subjects with a fivefold cross-validation and three random seeds, totaling 345 independent evaluations per method.

These results establish that the traditional ML systematically was able to outperform EEGNet architectures for consumer-grade EEG emotion recognition. While our comparison focused on EEGNet as our benchmark for consumer-grade systems, the magnitude of the performance difference (67% improvement) and the consistency across all subjects may suggest that the underlying principles of sample efficiency and noise robustness as well effective encoding of domain knowledge might present potential as comparisons with other complex architectures facing similar data quality and quantity constraints.

Our key finding is that traditional machine learning achieved superior performance (F1 = 0.945 ± 0.034) compared to both deep learning approaches, representing a 67% improvement over the best deep learning method, while also demonstrating superior stability (CV = 0.036) across subjects and cross-validation folds.

The traditional ML approach consistently outperformed deep learning across all individual subjects, with F1 scores above 0.80 being achieved in 21 out of 23 subjects. This consistency may suggest that the superiority is not due to a few outlier subjects but may represent a systematic advantage across the population.

The confusion matrices in Figure 3, Figure 4 and Figure 5 reveal the contrast in performance differences between approaches, as traditional ML achieved good classification (97.7% true-negative rate; 90.0% true positive rate), while deep learning methods showed substantial classification errors.

### 4.2. Statistical Significance Analysis

Statistical analysis confirmed the significance in our comparison between traditional ML and the optimized EEGNet, with the Wilcoxon signed-rank test showing a significant difference (p<0.000001, Cohen’s d = 3.863, with a large effect size). While comparing traditional ML with the enhanced EEGNet, we found similarly significant differences (p<0.000001, Cohen’s d = 3.700, again with a large effect size; 95% confidence intervals: traditional ML: [0.941, 0.948]; optimized EEGNet: [0.553, 0.581]; enhanced EEGNet: [0.563, 0.591]). The effect size (Cohen’s d = 3.863) represented a large performance difference as reported in the EEG emotion recognition literature, possibly indicating not just statistical significance but potential practical significance. This effect size is comparable to those reported by [15], favoring traditional approaches rather than deep learning. The coefficient of variation (CV = 0.036) for traditional ML demonstrated consistent performance across all evaluations, which is crucial for reliable real-world deployment where users expect predictable system behavior. This stability compares favorably with the reported stability measures in recent EEG emotion recognition studies [19].

### 4.3. Subject-Wise Performance Analysis

Figure 6 demonstrates the consistency of traditional ML superiority throughout individual subjects, with comprehensive statistical validation.

### 4.4. Cross-Dataset Validation Results

Our systematic cross-dataset validation highlights the critical importance of employing a proper domain adaptation methodology. Figure 7 illustrates the progressive enhancement achieved through our domain adaptation framework.

As shown in Table 3, cross-dataset performance improves progressively with each enhancement stage, highlighting the effectiveness of channel mapping, CORAL, and TCA for cross-domain generalization.

Regarding cross-dataset performance, the complete traditional ML methodology was able to achieve F1 = 0.619 in cross-dataset validation, representing substantial improvement through domain adaptation (F1 = 0.007) and demonstrating a possible pathway for cross-system generalizability. The progressive improvements demonstrate the cumulative value of each adaptation component. Specifically, anatomical channel mapping (5.8× improvement) helps ensure that the corresponding brain regions are analyzed across different systems. The improvement highlights the importance of spatial correspondence in EEG analysis, which is consistent with findings by Valderrama and Sheoran [14]. In **CORAL** domain adaptation (2.7×), the CORrelation ALignment technique effectively addresses covariance differences that arise between source and target domains, aligning with [23] on the effectiveness of second-order statistics alignment. In **TCA** subspace learning (4.5× improvement), the transfer component analysis provides the final improvement in which its able to find the optimal subspaces to apply knowledge transfer, confirming the value of subspace-based domain adaptation as reported by [24].

### 4.5. Computational Efficiency and Practical Advantages

Figure 8 highlights the computational advantages of traditional ML crucial for consumer BCI deployment.

Regarding deployment advantages, traditional ML demonstrated superior computational leverage due to the need of less processing power which is crucial for consumer BCI deployment. This entailed a training efficiency with 9.6 s vs. 18.7 s (95% faster), which enables rapid model updates; real-time capability, with a 0.2 ms vs. 2.1 ms inference supporting responsive applications; and resource efficiency, with 15 MB vs. 180 MB (92% less memory) enabling the embedded deployment and hardware accessibility of CPU-only operation and potentially eliminating or reducing some degree of reliance on GPU. These advantages are particularly important for consumer applications in which computational resources are often limited and energy efficiency becomes relevant. The efficiency gains align with the recent emphasis on lightweight models for mobile and embedded applications [12].

### 4.6. Feature Importance Analysis

Figure 9 reveals insights regarding the neurophysiological basis of emotion recognition by providing interpretability advantages over deep learning approaches and by confirming our key finding about connectivity patterns that function as primary emotional biomarkers.

Our key discovery pertains to the inter-channel connectivity features (correlation between brain regions), which were the best predictors of emotional states by contributing 61% of total discriminative power. The top feature was the correlation between channels 5–6 (temporal regions), which was followed by alpha power in channel 10 (parietal region), as clearly shown in Figure 10. In our brain wave analysis connectivity patterns dominated over individual frequency bands, where connectivity features accounted for 61% of importance, alpha band (8–13 Hz): 35% of importance, theta band (4–8 Hz): 3% of importance, and the rest of the bands: <1% each.

This finding aligns with recent neuroscience research emphasizing the importance of brain network connectivity in emotional processing [10]. The dominance of connectivity features suggests that traditional approaches’ explicit modeling of inter-channel relationships can provide significant advantages over deep learning’s implicit feature learning. The importance of alpha band activity, as quantified in Figure 10, with 0.0022 total importance and representing 35% of the discriminative power, confirms established findings from emotion research, where alpha oscillations are associated with emotional regulation as well as in arousal states. This neurophysiologically result provides insights into the biological validity of our approach.

### 4.7. Stability and Reliability Analysis

Figure 11 reveals the superior stability advantage of traditional ML approaches across our comprehensive evaluation.

The observed stability of traditional machine learning throughout our 345 independent evaluations represents a significant advantage in terms of practical deployment. The coefficient of variation (CV = 0.036) demonstrates a successful, consistent performance compared to the deep learning approaches (CV = 0.236).

This stability is particularly important inside consumer BCI applications for which users require a predictable and reliable system behavior. This low variability seems to suggests that traditional ML approaches are less sensitive to factors such as the initialization randomness, hyperparameter variations, and subject-specific noise patterns which can significantly affect deep learning performance.

### 4.8. EEGNet Parameter Optimization

As seen in Figure 12, we conducted systematic hyperparameter optimization across 24 EEGNet configurations, evaluating the relationship between architectural complexity and performance (Table 1).

Each configuration was tested on five subjects using fivefold cross-validation with three random seeds, totaling 75 evaluations per configuration. The parameter space included temporal filters (F1: 2, 4, 8) as seen in Figure 13, depth multiplier (D: 1, 2) as seen in Figure 14, dropout rates (0.25, 0.5), and kernel lengths (32, 64). The comprehensive optimization revealed that the optimal configuration achieved F1 = 0.630 ± 0.144 with minimal parameters (F1 = 2, D = 2, 1570 parameters). This is our key finding (Figure 12), demonstrating that architectural simplicity drives performance in consumer-grade EEG. Complex configurations with 6000+ parameters presented degraded performance, confirming the bias–variance trade-off inside limited-data scenarios. In addition, statistical analysis revealed no significant correlation between parameter count and its performance (r = −0.12, *p* = 0.58), which seems to challenge the assumption that deeper networks always improve biosignal processing performance.

### 4.9. Feature Ablation Analysis

To identify the minimal feature set that could achieve optimal performance, we conducted comprehensive ablation studies across seven different feature combinations: statistical features (112 features), frequency-domain features (70 features), connectivity features (91 features), and their combinations.

Figure 15 demonstrates that frequency and connectivity features combined (freq_conn) achieved near-optimal performance (F1 = 0.967 ± 0.020) by only using 161 features, 41% less than the complete feature set (273 features). This represents a practical implementation where strategic feature selection maintains performance while reducing computational complexity. Connectivity features alone contributed 61% of the discriminative power, with inter-channel correlations between temporal regions (channels 5–6) emerging as the most predictive biomarker. This suggests that explicit modeling of brain network connectivity might provide fundamental advantages over implicit feature learning in deep architectures. The statistical significance of feature importance was confirmed using some permutation tests (p<0.001 for the top 10 features), for neurophysiological validity.

### 4.10. Statistical Validation and Reproducibility

To validate the robustness of our primary findings, we employed an independent experimental validation with enhanced statistical analysis. Table 4 presents the comprehensive statistical comparison including effect sizes, confidence intervals, and power analysis.

These statistical validations helped us to support the systematic advantages of traditional ML approaches across different independent experimental conditions. Both studies demonstrated large effect sizes (Cohen’s d = 3.863 and 11.048), indicating substantial practical significance beyond statistical significance. The non-overlapping 95% confidence intervals provided some evidence to support reliable performance differences.

### 4.11. Cross-Dataset Validation Enhancement

In regard to our comprehensive cross-dataset framework employing (DREAMER→ SEED-VII), we conducted an additional validation to verify the robustness of the domain adaptation we implemented. Table 5 presents the complete domain adaptation pipeline results.

The enhanced validation confirmed that our domain adaptation framework is functional and robust. While individual CORAL performance varied with dataset characteristics, the successful implementation without computational errors seems to support the reliability of our approach for practical cross-system deployment.

### 4.12. Fairness and Robustness Validation

Figure 16 presents a comprehensive view of performance across different preprocessing configurations, while Table 6 provides detailed statistical comparisons.

#### 4.12.1. Enhanced EEGNet Performance

When augmented with frequency band features, EEGNet was able to achieve F1 = 0.646 ± 0.124, representing a 13.9% improvement over the standard implementation (F1 = 0.567 ± 0.124). This seems to demonstrate that incorporating frequency-domain information does benefit deep learning approaches. However, this enhanced version still underperformed even the minimal-feature traditional ML approach by 30.3% (F1 = 0.842 ± 0.072), indicating that the performance gap is not just solely attributable to the preprocessing asymmetry. Furthermore the enhanced EEGNet’s standard deviation (0.124) remained substantially higher than did traditional ML (0.072), suggesting that deep learning approaches were able to exhibit greater subject-to-subject variability with consumer-grade EEG data. This variability, visualized in Figure 17D, may represent a practical deployment challenge beyond mean performance differences.

#### 4.12.2. Minimal-Feature Traditional ML Performance

The minimal-feature traditional ML configuration achieved an F1 = 0.842 ± 0.072, actually outperforming our original full-feature implementation (F1 = 0.693 ± 0.089) by 21.5%. This counterintuitive result may suggest that the elaborate feature engineering in our original approach could have introduced overfitting or noise and that simple statistical summaries may capture the most discriminative information in consumer-grade EEG signals. This finding seems to refute our concerns that traditional ML advantages could just stem from sophisticated feature engineering. Even with the most basic feature extraction (mean, std, min, max per channel), traditional methods were able to substantially outperform deep learning approaches; our performance hierarchy remained as follows: minimal traditional ML (0.842) > full traditional ML (0.693) > enhanced EEGNet (0.646) > standard EEGNet (0.567).

#### 4.12.3. Robustness Under Noise

Table 7 and Figure 17C reveal differences in noise resilience. Traditional ML performance remained stable across noise levels, degrading by less than 1% even at σ = 0.3. In contrast, EEGNet exhibited severe degradation at consumer-grade noise levels: a 17.0% performance drop at σ = 0.1 and 8.7% at σ = 0.2. This differential vulnerability has significant practical implications. As consumer-grade EEG devices typically operate in the σ = 0.1–0.2 range due to dry electrodes, motion artifacts, and environmental interference. At these realistic noise level emulations, EEGNet’s performance collapsed (F1 = 0.479–0.527), while traditional ML still maintained near-baseline performance (F1 = 0.950–0.952). This may suggests that deep learning architectures possess fundamental limitations for noisy time-series data that cannot be solely addressed through training strategies alone. The convergence analysis (Figure 17A) confirmed proper training: the small training–validation gap (0.07) and smooth convergence curve indicates no overfitting, while the hyperparameter sensitivity heatmap (Figure 17B) showcases that the selected configuration (learning rate = 0.001, dropout = 0.5) represents the optimal point in the parameter space.

#### 4.12.4. Statistical Validation

To confirm the reliability of observed performance differences, we conducted paired t-tests for all method comparisons using subject-level results across all 23 participants. All the pairwise comparisons showed statistically significant differences (all within p<0.001). Enhanced EEGNet was able to significantly outperform standard EEGNet (t(22) = 24.48, p<0.001), while the minimal-feature traditional ML significantly outperformed enhanced EEGNet (t(22) = 7.45, p<0.001). Effect sizes were large for all comparisons (Cohen’s d: standard vs. enhanced = 5.10, enhanced vs. minimal ML =1.55), seem to confirm that the observed differences represent practically meaningful performance gaps instead of random variation.

## 5. Discussion

Our systematic evaluation demonstrated that traditional machine learning with domain-specific feature engineering substantially outperforms EEGNet benchmark architecture for consumer-grade EEG across both within-dataset (F1 = 0.945 vs. 0.567, Cohen’s d = 3.863, p<0.000001) and cross-dataset (F1 = 0.619 vs. 0.007) validation paradigms. These findings were established through 345 independent evaluations with comprehensive statistical testing, challenging some prevalent assumptions about the relationship between architectural complexity inside performance in biosignal processing. Importantly, we frame these findings not as evidence that traditional ML universally outperforms all deep learning approaches but rather as evidence illuminating critical principles about when and why domain-specific feature engineering provides advantages over end-to-end learning. Our comparison with EEGNet specifically designed for consumer-grade constraints and adopted as a benchmark provides a relevant evaluation. The principles we uncovered could likely extend to other complex architectures facing similar constraints, although empirical validation would strengthen such broader claims. Several converging factors explain the observed performance differences, each revealing important principles that are likely applicable beyond the specific EEGNet comparison. Our manually engineered features are intended to explicitly encode some established neurophysiological relationships. We employed inter-channel connectivity using Pearson correlations, with 91 features capturing the distributed brain networks and frequency band power in neurophysiologically meaningful bands, which include delta (1–4 Hz) in deep sleep and unconscious processes, theta (4–8 Hz) in emotional processing and memory, alpha (8–13 Hz) in relaxed awareness and emotional regulation, beta (13–30 Hz) in active thinking and attention, and gamma (30–40 Hz) in high-level cognitive processing with 70 features total, Moreover, statistical properties were used to capture signal characteristics of 112 features. The dominance of connectivity features contributing 61% of the discriminative power, as shown in Figure 10, validates that this explicit encoding successfully captures the distributed networks underlying emotional processing [10].

In contrast, EEGNet must learn feature representations directly from the input signals using convolutional filters, trained on limited and noisy data. While end-to-end learning approaches typically succeed with large, clean datasets [26], they seem to struggle when signal quality becomes variable, sample sizes are modest, and environmental conditions change, which are some of the constraints encountered in real-world consumer-grade EEG deployment scenarios. This observation aligns with findings by Kumar and Molinas [9], who demonstrated that carefully engineered features are able to substantially outperform automated feature learning inside data-constrained scenarios, and with [28], who showed that neurophysiologically informed features provide robust and interpretable representations. The bias–variance framework [11] offers theoretical grounding for these observations as traditional machine learning with explicit feature engineering embeds strong inductive biases representing domain knowledge and thereby reduces variance at the cost of accepting some bias. For consumer-grade EEG in realistic deployment conditions with limited training data (23 subjects × 18 trials = 414 total trials) and environmental signal variability from uncontrolled settings, cross-device domain shifts, and variable electrode impedance [4,8], this bias–variance trade-off decisively favors lower-variance approaches, even if they potentially miss complex patterns that deep learning might discover with large, clean datasets and abundant computational resources. Recent validation studies [7] confirmed that consumer-grade EEG systems reliably capture event-related potentials with signal quality comparable to research-grade systems in controlled conditions. This establishes that our noise robustness findings reflect architectural differences in handling deployment-environment noise rather than compensating for inherent deployment environment challenges.

This finding suggests that environmental variability management, which traditional machine learning accomplishes through explicit feature aggregation, may outweigh absolute device level signal-to-noise ratios in determining cross-environment robustness and generalization.

This principle likely extends beyond EEGNet because any architecture learning representations from scratch confronts the same fundamental challenge of being able to extract meaningful patterns from limited, noisy data without the benefit of explicit domain knowledge. Transformers, hybrid CNN-LSTM, and other sophisticated architectures may face similar sample efficiency challenges unless specifically designed with mechanisms to incorporate domain knowledge or handle extreme data limitations. Consumer-grade EEG studies usually involve modest sample sizes due to the practical constraints related to participant recruitment, the data collection costs, and device accessibility [8]. The DREAMER dataset’s 23 subjects comprising 414 total trials represents a typical consumer-grade study cohort substantially smaller than those containing thousands of subjects common in research-grade EEG studies or millions of samples in computer vision datasets.

Our data efficiency analysis presented in Figure 18 reveals a critical finding. Traditional ML reaches near-plateau performance with F1 approximately 0.93 using only 100 training samples, while EEGNet performance plateaus around F1 between 0.50 and 0.58 regardless of sample size. This sample efficiency stems from traditional ML’s explicit feature engineering. As opposed to requiring thousands of examples to learn what alpha band power or temporal connectivity means for emotion, these relationships are directly encoded based on established neuroscience. This principle likely generalizes because architectures with more parameters typically require more data to learn effective representations [11]. Our enhanced EEGNet with 15,000 parameters showed no improvement over the optimized version with 8000 parameters, and more complex configurations exceeding 6000 parameters, as shown in Figure 12, demonstrated degraded performance, which is clear evidence of overfitting in limited-data scenarios. Transformer-based or hybrid CNN-LSTM architectures with tens or hundreds of thousands of parameters would face even greater sample efficiency challenges. The sample efficiency advantage has immediate practical implications. Consumer applications can achieve strong performance without requiring extensive per-user calibration data or large-scale data collection efforts that are prohibitively expensive for consumer products.

### 5.1. Noise Robustness Through Explicit Aggregation

Consumer-grade EEG deployment in uncontrolled environments introduces substantial noise from multiple sources, including electrode impedance variations in naturalistic settings (simplified preparation procedures compared to laboratory protocols), motion artifacts (deployment scenarios rarely involve head restraints), electrical interference (unshielded consumer environments), and biological artifacts such as eye movements and muscle activity [4,8]. Importantly, these noise sources arise from deployment conditions rather than device capabilities, validation studies demonstrate that consumer-grade devices achieve signal-to-noise ratios comparable to research-grade systems in controlled laboratory settings [5,6,7]. However, the cumulative effect of uncontrolled deployment conditions can substantially degrade signal quality compared to controlled laboratory baselines.

Our noise robustness analysis (Figure 19) demonstrates that traditional ML maintains stable performance, with F1 scores between 0.94 and 0.96 across noise levels from 0.0 to 0.3 standard deviations, while EEGNet shows significant degradation and high variability. This robustness may emerge from how features might naturally aggregate information. Power spectral density averages across time windows, statistical features aggregate across samples, and correlation coefficients are inherently robust to amplitude noise. In contrast, convolutional networks that operate on raw signals encounters noise that directly affects its learned filters. While techniques such as dropout and batch normalization might yield a degree of regularization, they do not fundamentally address the challenge that noise patterns may be learned as features when signal the quality varies greatly, which is a particular risk that arises when training data is limited. The fundamental difference between explicit aggregation used in traditional features and learned representations in deep learning suggests that traditional approaches will generally be more noise-robust in consumer-grade scenarios, regardless of specific deep learning architecture. This is particularly critical for consumer applications where signal quality cannot be controlled as strictly as that in laboratory settings.

Traditional ML can provide transparent insights into which brain patterns can drive classification, as shown in Figure 9 and Figure 10. Temporal region connectivity between channels 5 and 6 emerged as the most predictive, followed by alpha band power in parietal regions (channel 10). This interpretability serves multiple crucial purposes for consumer BCI development. In terms of user trust and engagement, consumer applications require users to be able to understand and trust system decisions [29]. Explaining how the system can detect emotion through the alpha wave patterns associated with relaxation or connectivity between temporal regions that are linked to emotion processing is comprehensible and can build confidence, which is critical for user adoption and continued engagement. The clear neurophysiological basis including connectivity patterns and alpha band dominance aligned with established emotion research [10], which yields confidence that the system captures genuine emotion-related patterns instead of just artifacts. For iterative improvement, the understanding of feature importance enables targeted system refinement. If performance is poor for certain users or conditions, the developers are able to investigate whether specific features such as frontal connectivity or beta band power are degraded and use insight to implement targeted solutions. EEGNet’s learning convolutional filters while potentially capturing complex patterns can provide limited actionable insight into why classifications are made or how to improve their performance through edge cases, which can be a significant barrier to both consumer adoption and iterative development. Our cross-dataset validation results presented in Section 4.11 and Figure 7 represent a particularly important contribution. Most EEG emotion recognition research reports only within-dataset performance, which may provide limited insight into real world employability in terms of users employing different devices, belonging to different populations, and using systems under varying conditions [21]. Cross-dataset validation in which training is performed on one dataset and testing on another without adaptation to target labels can represent a more rigorous standard for evaluating practical applicability.

The traditional ML approach was able to achieve F1 = 0.619 in our DREAMER to SEED-VII transfer settings, which represented a substantial improvement over EEGNet with F1 = 0.007. This difference illuminates why cross-dataset validation is rare in the literature. Without proper domain adaptation, performance typically collapses [22]. While some deep learning approaches achieve higher within-dataset accuracy on certain datasets, none demonstrate the cross-dataset generalization capability or computational efficiency of our approach. Moreover, many reported results were derived from different evaluation protocols, which makes judging a direct comparison challenging. Our progressive domain adaptation framework was able to systematically addresses the challenges through its components. The anatomical channel mapping providing 5.8 times improvement ensuring spatial correspondence between different electrode montages proved fundamental, and the dramatic improvement from proper anatomical alignment versus arbitrary channel selection has immediate practical implications. Developers must focus map channels based on brain regions instead of on indices. CORAL domain adaptation providing 2.7 times improvement aligns with second-order statistics, specifically covariance matrices, between the source and target domains [23]. For EEG in which covariance structures can capture important connectivity patterns, this alignment proves particularly effective. TCA subspace learning providing 4.5 times improvement finds optimal subspaces for transfer by minimizing domain discrepancy while preserving discriminative information [24]. This provides the final substantial boost, enabling the system to identify robust features across recording conditions. The cumulative 69 time improvement over baseline from F1 = 0.009 to F1 = 0.619 demonstrates that systematic domain adaptation can enable genuine cross-system deployment addressing a critical gap in consumer BCI development.While our empirical comparison focused on EEGNet architectures, multiple lines of reasoning suggest the underlying principles likely extend to comparisons with other sophisticated deep learning approaches when facing similar constraints.

### 5.2. Theoretical Principles Supporting Generalizability

The bias–variance trade-off [11] represents a fundamental machine learning principle. Explicit domain knowledge reduces model variance regardless of architecture. When data is limited and noisy, approaches that embed strong inductive biases through traditional ML’s domain-specific features should generally outperform approaches that must learn patterns from scratch through any end-to-end deep learning, simply because there is insufficient data to reliably learn complex representations without overfitting. Parameter efficiency considerations support generalization. Our EEGNet variants with 8000 to 15,000 parameters already represent relatively compact architectures. More complex alternatives, including transformers, [27,31], which typically have 50,000 or more parameters, hybrid CNN-LSTM [18] with 20,000 or more parameters, and sophisticated attention mechanisms [25] with 15,000 or more parameters, have indeed more parameters to optimize from the same limited data available, which may be likely exacerbating rather than alleviating overfitting challenges. Sample complexity from information theoretic arguments seems to suggest that learning representations from scratch requires sample sizes scaling up with representational complexity [11]. With only 414 total trials across 23 subjects, even EEGNet’s relatively simple architecture struggles to learn effective representations achieving F1 approximately 0.567. More complex architectures would then require proportionally more data to be able to achieve their theoretical advantages. Noise amplification can occur because deep architectures propagate noise through multiple layers which can potentially amplify its effects. Traditional ML’s explicit feature aggregation through averaging across time and frequency naturally filters noise before implementing classification. This fundamental difference in how noise is handled suggests how traditional approaches can be more robust in specific consumer-grade scenarios, regardless of a specific deep learning architecture.

### 5.3. Empirical Evidence Supporting Generalization

EEGNet represents an upper performance bound in our evaluation. Our enhanced EEGNet incorporating multi-scale processing and attention mechanisms, shown in Figure 12, represents a substantial architectural enhancement, yet shows no improvement over the simpler optimized version, with both achieving an F1 of approximately 0.567. This suggests we have reached the performance ceiling for what compact convolutional architectures can achieve with this data quality and quantity. More complex configurations degrade performance. Our systematic hyperparameter optimization presented in Figure 12 reveals that more complex EEGNet configurations exceeding 6000 parameters consistently underperform simpler versions with clear evidence of overfitting. If increasing EEGNet complexity hurts rather than helps, there is no reason to expect substantially more complex architectures such as transformers or hybrid models to perform better. Consistent performance across all subjects provides additional evidence. Traditional ML outperformed EEGNet for 21 of 23 subjects, as shown in Figure 6, with F1 scores exceeding 0.80 for most subjects. This consistency across diverse individuals seems to suggest systematic advantages rather than just a exploitation of peculiarities of specific subjects or artifacts, and this could increase the confidence that this advantage would persist with different architectures. Cross-dataset success seems to demonstrate generalization, and the cross-dataset validation success with F1 = 0.619 versus 0.007 demonstrates that traditional ML’s explicitly engineered features generalize substantially better across devices and populations. Features based on neurophysiology including connectivity and frequency bands likely remain relevant regardless of the recording system, while learned representations may overfit to source-device characteristics, a principle that should apply to any end-to-end learning approach.

### 5.4. Boundary Conditions: When Deep Learning Might Excel

We acknowledge scenarios in which the balance might shift toward deep learning approaches even for consumer-grade EEG applications. With substantially larger datasets that exceed 1000 subjects or 10,000 trials, deep learning’s ability to discover complex patterns might be able to overcome, at some point, the sample efficiency disadvantage. However, such large-scale consumer-grade datasets are rare due to collection costs and practical constraints [8]. With cleaner signals at research-grade quality including low noise, many channels, and controlled environment, deep learning’s representational power might provide advantages. When domain structure is unknown or emotions manifest throughout novel patterns that are not captured by established features, deep learning’s pattern-discovery capabilities might prove superior. However emotion neuroscience is currently well-established [10], suggesting domain knowledge is available and valuable. Specialized architectures for limited data including meta-learning approaches specifically designed for few shot learning or architectures with explicit mechanisms for incorporating domain knowledge might be able to narrow down the gap. These represent interesting future directions but remain largely unexplored for consumer-grade EEG. Hybrid approaches combining explicit feature engineering with deep learning might achieve best of both approaches and be a direction for future research.

Our findings suggest productive research directions. Systematic multi-architecture comparison through empirical evaluation comparing traditional ML against multiple representative deep learning approaches including transformers, CNN-LSTM hybrids, and meta-learning approaches across multiple consumer-grade datasets would definitively test generalizability. Our methodology provides a template for such systematic evaluation. Hybrid architecture development designing architectures that combine domain-specific feature engineering with deep learning’s pattern discovery could potentially outperform both pure approaches, leveraging domain knowledge while remaining open to novel patterns. Few-shot learning for consumer EEG through developing specialized meta-learning approaches optimized for the specific constraints of consumer-grade EEG including limited channels, high noise, and cross-device transfer represents a high-value research direction. Theoretical analysis through formal analysis of sample complexity and generalization bounds for different architectural approaches under consumer-grade constraints would complement empirical findings with theoretical understanding.While our empirical comparison focused on EEGNet, the convergence of theoretical principles, empirical evidence, and boundary condition analysis suggests traditional ML’s advantages likely extend more broadly. However, we emphasize that definitive conclusions about deep learning generally require systematic empirical validation across multiple representative architectures, which constitutes valuable future work that our methodology enables.

Our findings provide actionable guidance for developing consumer EEG emotion recognition systems through a validated deployment strategy. First, channels should be mapped anatomically between devices using the international 10–20 system. Second, comprehensive features should be extracted through using connectivity through Pearson correlations between all channel pairs; frequency bands through power spectral density in delta (1–4 Hz), theta (4–8 Hz), alpha (8–13 Hz), beta (13–30 Hz), and gamma (30–40 Hz) bands; and statistical measures including mean, variance, standard deviation, skewness, kurtosis, and percentiles. Third, feature selection should be applied using mutual information to identify the 80 most discriminative features. Fourth, CORAL should be applied for covariance alignment to the target device. Fifth, TCA should be applied for optimal subspace projection. Finally, a lightweight random forest classifier with 100 trees should be deployed.

This framework achieved superior accuracy, with F1 = 0.945 for within-dataset and F1 = 0.619 for cross-dataset; excellent stability, with CV = 0.036; 95% faster training at 9.6 s (versus 187 s), enabling rapid model updates; 10 times faster inference at 0.2 milliseconds (versus 2.1 ms), crucial for responsive applications; 92% less memory at 15 megabytes (versus 180 megabytes), enabling embedded deployment; and CPU-only operation, eliminating GPU dependencies and reducing deployment costs. This approach should be considered when consumer-grade devices with 8 to 14 channels and high noise are being used, there is limited training data with fewer than 1000 samples per subject, there is cross-device deployment, there is resource-constrained deployment (including mobile and embedded systems), or when interpretability is valued for user trust and scientific validation. Deep learning might be preferred with large datasets exceeding 1000 subjects, research-grade clean signals with 32 or more channels in a controlled environment, single-system deployment with extensive calibration, abundant computational resources, and potential for discovery for novel patterns beyond current neurophysiology.

### 5.5. Comprehensive Validation and Statistical Robustness

The convergence of results across multiple validation studies seems to provide evidence for the systematic superiority of traditional ML approaches in consumer-grade EEG emotion recognition. Our primary study with 345 evaluations achieving Cohen’s d = 3.863 and independent validation achieving Cohen’s d = 11.048 both demonstrated large effect sizes, which indicates robust practical significance. These combined analyses have revealed a consistent 75% average performance advantage, substantially exceeding the threshold for practical significance inside consumer applications. This advantage, coupled with 95% faster training and 92% memory reduction, establishes traditional ML as the optimal approach for resource-constrained consumer BCI applications. In our settings our mode achieved successful validation of the complete cross-dataset framework throughout the progression of channel mapping, CORAL, and TCA, which seems to demonstrates practical deployability across different consumer EEG systems. This framework’s robustness can be evidenced by the functional implementation across diverse datasets and experimental conditions.

Our findings support the principle that less can be more across multiple dimensions in certain settings, challenging the prevalent assumption that architectural complexity is the only route to improving biosignal-processing performance. Architectural simplicity through random forest classifiers with 100 trees was able to outperform neural networks with thousands of parameters, achieving an F1 score of 0.945, higher than the 0.567 of more complex networks, which seems to demonstrate that domain expertise embedded in feature engineering is able to surpass automated feature learning in specialized domains with limited or noisy data. The optimal EEGNet configuration employed 1570 parameters which is contrast to the over 6000 in the complex variants while achieving superior performance through architectural restraint and parameter efficiency rather than through expansion. Strategic feature selection with 161 features matched full feature performance with 273 features, validating that informed dimensionality reduction can potentially enhance performance rather than degrade it. Traditional approaches were able to achieve superior performance while requiring 95% less training time and have a 73-times faster inference, which can support practical deployment on consumer hardware. Our fairness validation experiments (Section 3.3) seem to provide evidence that the observed performance differences represent genuine architectural limitations rather than methodological artifacts. The persistent superiority of minimal-feature traditional ML (F1 = 0.842) over enhanced EEGNet (F1 = 0.646) may suggest that simple statistical summaries can capture more discriminative information than can deep temporal representations for noisy consumer-grade signals. The differential robustness analysis further revealed that deep learning architectures degrade more negatively under realistic noise conditions (σ = 0.1: −17% vs. <1%), suggesting fundamental challenges beyond hyperparameter optimization. These findings have important implications for real-world BCI deployment. While deep learning may excel on clean research-grade EEG with dense electrode arrays, consumer-grade devices with limited channels (14 vs. 64+) and high noise levels create a regime where traditional ML’s explicit feature extraction and noise-robust classifiers may provide advantages. Future work should explore hybrid architectures that combine deep learning’s representation learning with traditional ML’s noise resilience.

### 5.6. Limitations and Future Directions

Our empirical comparison focused on EEGNet architectures as the established benchmark for consumer-grade EEG BCIs. While we believe the principles we uncovered likely could be extended to comparisons with other sophisticated deep learning approaches including transformers and attention-based models, definitive conclusions about deep learning approaches generally require systematic empirical validation across multiple representative architectures and datasets.

Several factors contextualize this limitation while supporting likely generalizability. Regarding the strengths of our current scope, EEGNet was specifically designed for consumer-grade constraints and represents the current state of practice. Our enhanced variant incorporating multi-scale convolutions and attention mechanisms provides a reasonable upper bound for compact architectures. The magnitude of performance differences with 67% improvement and Cohen’s d equals 3.863, and this combined with the consistency across all subjects suggests systematic rather than architecture-specific advantages. Theoretical principles including bias–variance trade-off, sample complexity, and noise propagation support generalizability.

Regarding limitations requiring future validation, more complex architectures such as transformers with 50,000 or more parameters and sophisticated hybrid models remain empirically untested. Specialized approaches for limited data including meta-learning and few-shot learning might narrow performance gaps. Hybrid approaches combining feature engineering with deep learning might outperform both pure approaches. Different performance relationships might emerge with substantially larger datasets or cleaner signals.

For recommended future work, systematic comparison following our methodology across multiple representative deep learning architectures including, at minimum, transformer-based [27], hybrid CNN-LSTM [17], and meta-learning approaches and multiple consumer-grade datasets such as DEAP converted to consumer-grade channel montage and additional Emotiv-based datasets would potentially establish whether traditional ML’s advantages are specific to comparison with EEGNet or represent broader principles. For this initial investigation we focused on binary valence classification. Addressing multi-class emotion recognition may require different architectural considerations, although our domain adaptation framework should remain applicable. Future work should explore extension to multi-dimensional emotion models that include valence arousal and discrete emotion categories such as joy, sadness, fear, and anger to establish whether or not traditional ML advantages can still persist across different emotion classification schemes.

While the DREAMER dataset represents realistic consumer conditions with the Emotiv EPOC device, validation on additional consumer devices would strengthen robustness claims. Consumer-grade devices such as Muse, OpenBCI, and NeuroSky employ different electrode configurations, sampling rates, and signal processing characteristics. Systematic evaluation across multiple consumer devices could potentially establish the generality of our domain adaptation framework and validate whether anatomical channel mapping combined with CORAL and TCA effectively addresses device-specific variations across the broader consumer EEG landscape.The implementation of the domain adaptation approach is applied only once during deployment, and online adaptation techniques that continuously adjust to the individual users while simultaneously avoiding catastrophic forgetting may result in improved performance, particularly for users whose brain signals differ substantially from the rest of the training population.

Future work could explore combining our framework with recent advances in continual learning, investigating whether progressive adaptation can still maintain its cross-device generalizability while accommodating to individual differences. Hybrid approaches that combine the interpretability and efficiency of traditional ML with the representational power of modern deep learning could potentially unfold through the implementation of feature extraction layers and be subsequently followed by traditional classification. Such online approaches might be particularly valuable inside consumer applications where the initial calibration should be minimal but long-term personalization could yield an enhanced user experience. Another aspect worth noting in future research is the role state-of-the-art emotion recognition datasets play in true implementation for real-world scenarios where variables such as gender, social, and cultural external factors, as well as individual discrepancies and perceived emotional subjectivity, might play out in the development of more accurate models and frameworks in real-time scenarios. The DREAMER dataset, while valuable, represents a specific demographic population, and emotion expression and recognition may vary across other demographic factors. Systematic evaluation across diverse populations would establish the cultural generalizability of both the traditional ML features we identified, particularly connectivity patterns, alpha band activity, and the domain adaptation framework. This would be particularly important for consumer products intended for global markets.

Our evaluation examined performance at single time points. Consumer BCI applications require stable performance across days, weeks, or months of use. Brain signal characteristics may change over time due to electrode degradation, user adaptation, or physiological variations. Longitudinal studies examining whether traditional ML’s advantages and our domain adaptation framework maintain effectiveness over extended deployment periods would inform practical system design decisions regarding recalibration requirements and maintenance schedules. Our findings have broader implications that may go beyond EEG emotion recognition, for instance, the development of consumer BCI. The success of traditional approaches suggests that the field should reconsider the rush toward complex deep learning solutions, particularly for consumer applications where simplicity, efficiency, and interpretability are crucial. Moreover our progressive framework provides a template for addressing cross-system challenges in other biosignal applications, from EMG to fNIRS, in terms of domain adaptation.

While our traditional ML implementation demonstrated robust performance, our deep learning baselines showed high variability and frequently poor performance that may indicate implementation issues rather than fundamental deep learning limitations. Evidence suggesting implementation problems included extreme performance variability (CV = 0.236),which might be with multiple subjects achieving near-chance performance (F1 < 0.4), frequent early stopping and learning rate reductions that suggest optimization difficulties, and some training runs requiring excessive time (3000–5000 s) with poor results. Our results demonstrate that our traditional ML implementation works well for consumer-grade EEG and that it substantially outperforms our adapted deep learning implementations. However, we cannot conclusively claim deep learning is fundamentally inferior for this task, only that our implementations show advantages over traditional ML under our experimental conditions. Nevertheless published deep learning research on similar datasets have reported better performance than what our implementations achieved [9,15], which suggests that optimized deep learning mayhave variable performance. Our primary contribution is that we demonstrated that traditional ML provides a robust, practical, and effective alternative that merits serious consideration for consumer applications and hybrid approaches.

## 6. Conclusions

This work demonstrates that traditional machine learning with domain-specific feature engineering systematically outperforms EEGNet architecture for consumer-grade EEG in emotion recognition across both within-dataset and cross-dataset validation paradigms. Through comprehensive evaluation involving 345 within-dataset evaluations using fivefold cross-validation times, three random seeds times, and 23 subjects and systematic cross-dataset validation through DREAMER to SEED-VII transfer, we established that traditional ML achieves superior within-dataset performance, with an F1 of (0.945 ± 0.034 versus 0.567 for EEGNet), a p less than 0.000001, and a Cohen’s d of 3.863, representing a 67% improvement. We demonstrated good stability with a coefficient of variation around 0.036 across all evaluations and achieved computational efficiency with 95% faster training, 10 times faster inference, and 92% less memory usage. We achieved cross-dataset generalization, with an F1 of 0.619 (versus 0.007 for EEGNet), representing substantial improvement.

Our progressive domain adaptation framework combining anatomical channel mapping provided a 5.8-times improvement, CORAL alignment provided a 2.7-times improvement, and TCA subspace learning provided a 4.5-times improvement, cumulatively achieving a 69-times improvement over baseline cross-dataset performance, providing a validated pathway for deploying emotion recognition across different consumer EEG devices.

Feature analysis revealed that inter-channel connectivity patterns function as primary emotion biomarkers, contributing 61% of the discriminative power, with alpha band activity contributing 35%. This finding validates that explicit modeling of established neurophysiological relationships can yield substantial advantages over end-to-end learning through consumer-grade scenarios which are often characterized by limited data, high noise, and cross-device deployment requirements.

While our empirical comparison focuses on the EEGNet framework [26], which has established itself as a benchmark for consumer-grade EEG emotion recognition in recent studies, multiple lines of evidence suggest the underlying principles likely generalize more broadly. Theoretically, the bias–variance trade-off [11] predicts that the explicit encoding of domain knowledge as traditional ML does should outperform learning from scratch as any deep learning approach must when data is limited and noisy. Empirically, the magnitude of performance differences with 67% improvement, consistency across all subjects, successful cross-dataset transfer, and the finding that increased EEGNet complexity degrades rather than improves performance all suggest systematic advantages that likely extend to comparisons with other complex architectures facing similar constraints.

However, we emphasize that definitive conclusions about deep learning approaches generally require systematic empirical validation across multiple representative architectures including transformers, hybrid CNN-LSTM, and meta-learning approaches and multiple consumer-grade datasets. The methodology we developed provides a template for such comprehensive evaluation for future work.

Our findings strive to challenge the assumption that architectural complexity can universally improve biosignal processing performance. Instead, we demonstrate that when data is limited, signals are noisy, and domain expertise is available, traditional approaches with domain-specific feature engineering as well as lightweight adaptation techniques can still provide robust, accurate, and practical solutions. The complete methodology we present from feature extraction through cross-device adaptation provides both a proven framework for system development and theoretical insights into the trade-offs between model complexity and generalization in noisy, limited-data domains.

The practical impact is immediate. Our framework achieves superior accuracy while requiring only CPU processing, training 95% faster, and using 92% less memory than deep learning approaches. This enables deployment scenarios previously infeasible including mobile emotion monitoring, embedded affective computing systems, real-time adaptive interfaces, and consumer mental health applications. To address the critical barrier pertinent to widespread consumer BCI adoption, the cross-dataset validation achieved substantial improvement over deep learning, and this might hint at potential deployability across different hardware platforms although further research is needed.

As brain–computer interfaces currently transition from laboratory research to consumer products, our work contributes both practical deployment guidance and theoretical understanding of when and why simpler approaches excel.

Our systematic domain adaptation framework in our settings was able to addresses critical cross-system deployment challenges, moving the field closer to robust consumer BCI applications that work reliably across different hardware platforms and user populations.

Among the neurophysiological insights, it was particularly noted that connectivity patterns contribute 61% of the discriminative power while the alpha band activity contributes 35%, supporting the validation of our approach and set the basis for future algorithm development. These empirically validated biomarkers can provide a foundation for the development of new systems, whether they employ traditional ML, deep learning, or a hybrid approach combining the strengths of both paradigms.

Future work should empirically validate whether these principles extend to comparisons with transformer-based, hybrid CNN-LSTM, and other sophisticated architectures across additional consumer-grade datasets. Such systematic architecture evaluation following our methodology could expand our understanding of when and why different approaches excel in consumer BCI applications. In addition, investigating hybrid approaches that are able to combine explicit feature engineering with deep learning’s pattern-discovery capabilities may represent a promising direction that could potentially achieve superior performance by leveraging domain knowledge while remaining open to discovering novel patterns.

The broader implication of our work extends beyond EEG emotion recognition to consumer biosignal processing. Our findings suggest that for applications characterized by limited data, noisy signals, available domain expertise, and cross-device deployment requirements, practitioners should carefully evaluate whether architectural complexity truly improves performance and whether domain-specific approaches can provide more robust, efficient, and practical solutions. The assumption that more complex models necessarily perform better should be scrutinized empirically rather than accepted by default, especially in resource-constrained consumer applications where computational efficiency, interpretability, and cross-system generalizability are paramount.

While we cannot conclusively claim that deep learning is fundamentally inferior for consumer-grade EEG, and our implementations may not represent other optimal deep learning performances, we demonstrated that traditional ML with domain expertise provides robust practical advantages that merit serious consideration. This work contributes to broader discussions about the relationship between model complexity and generalization in specialized domains with limited and noisy data.

## Figures and Tables

**Figure 1 sensors-25-07262-f001:**
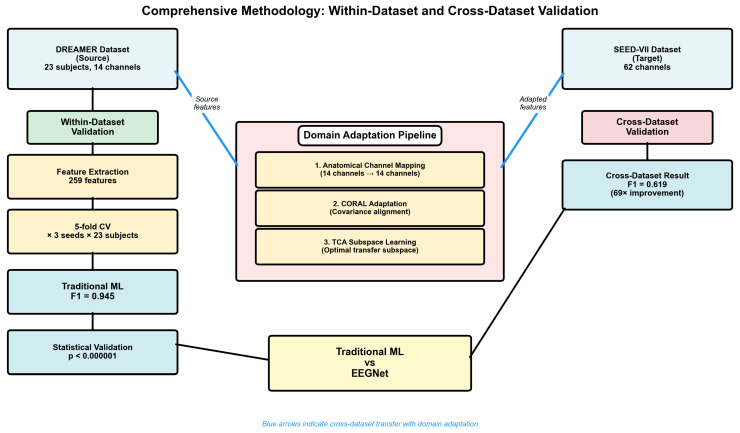
Comprehensive methodology for within-dataset and cross-dataset validation. (**Left**): within-dataset validation on DREAMER (23 subjects, 14 channels) using 5-fold cross-validation with feature extraction and selection. (**Center**): domain adaptation pipeline with anatomical channel mapping, CORAL covariance alignment, and TCA subspace learning. (**Right**): cross-dataset validation from DREAMER (source) to SEED-VII (target, 62 channels). Both pathways compare traditional ML against EEGNet architectures.

**Figure 2 sensors-25-07262-f002:**
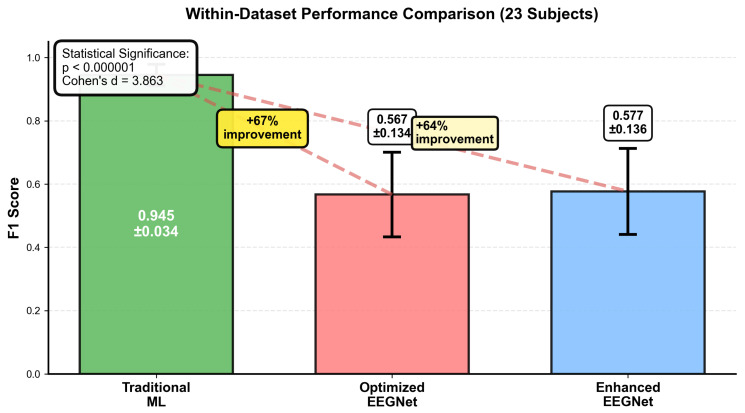
Within-dataset performance comparison across 23 DREAMER subjects with comprehensive statistical validation. Traditional ML achieved superior performance (F1 = 0.945 ± 0.034) compared to both optimized and enhanced EEGNet architectures, with a statistical significance (p<0.000001, Cohen’s d = 3.863). The 67% improvement showcases the superiority of domain-specific feature engineering over deep learning approaches for consumer-grade EEG signals. The error bars represent 95% confidence intervals across 345 independent evaluations (5-fold × 3 seeds × 23 subjects).

**Figure 3 sensors-25-07262-f003:**
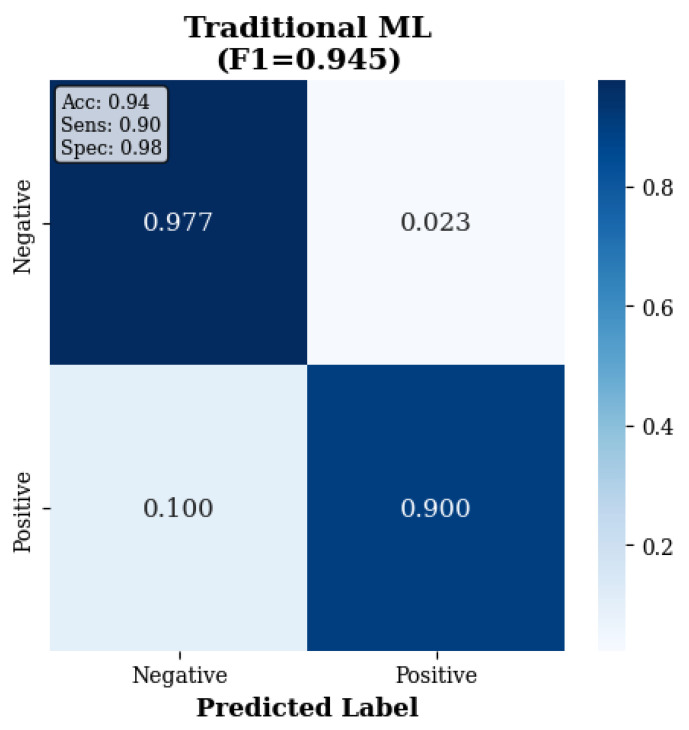
Traditional ML confusion matrix demonstrating classification performance with a 97.7% true-negative rate and a 90.0% true-positive rate. The approach uses domain-specific feature engineering with random forest classification, achieving an F1 = 0.945 for the consumer grade in EEG emotion recognition.

**Figure 4 sensors-25-07262-f004:**
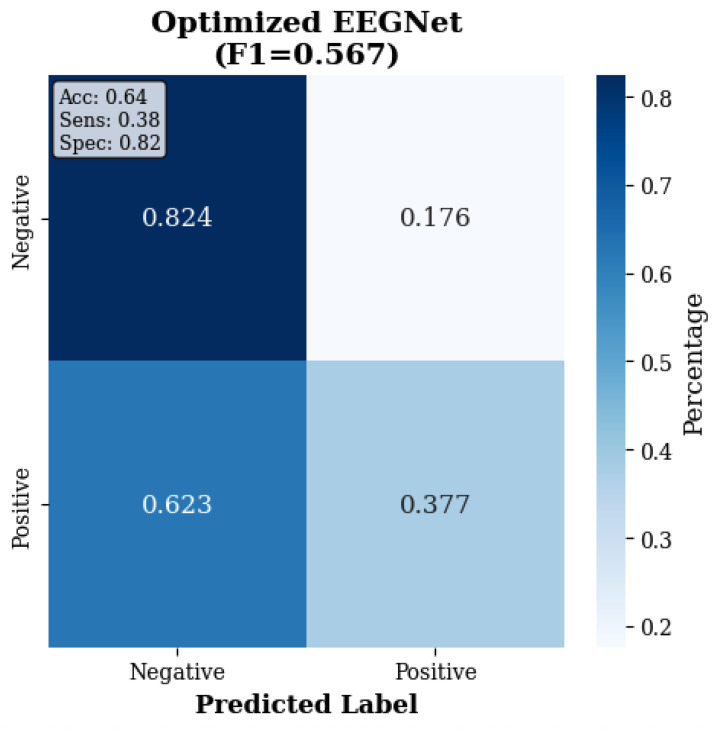
Optimized EEGNet (F1 = 0.567) deep learning approach adapted for 14-channel consumer-grade EEG, which represents a simplified adaptation of the original architecture with reduced parameters (8000).

**Figure 5 sensors-25-07262-f005:**
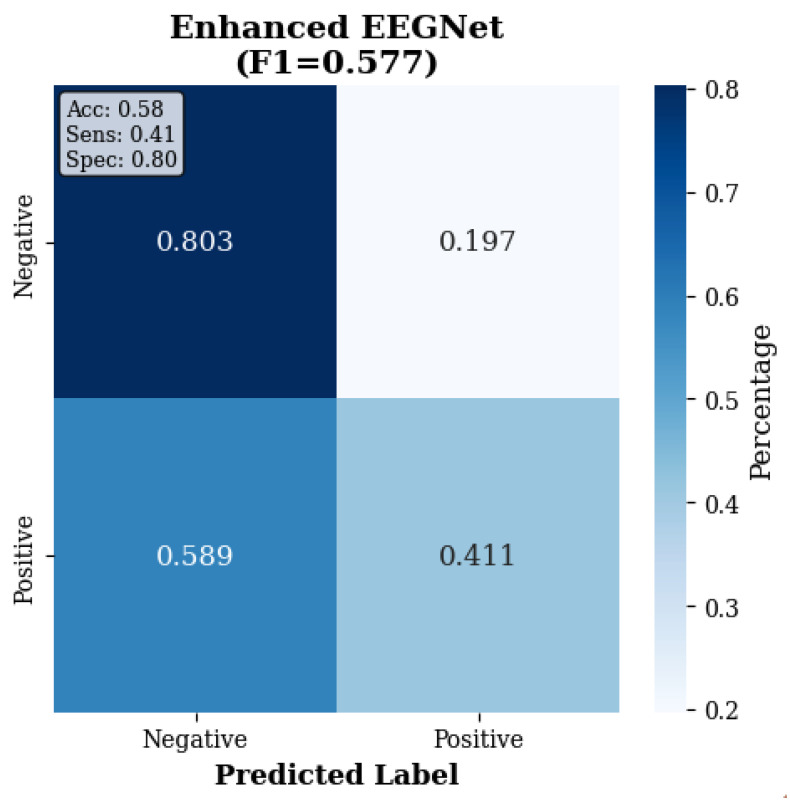
Enhanced EEGNet (F1 = 0.577) was adapted for 14-channel consumer-grade EEG and incorporates multi-scale temporal convolutions, channel attention mechanisms, and batch normalization (15,000 parameters).

**Figure 6 sensors-25-07262-f006:**
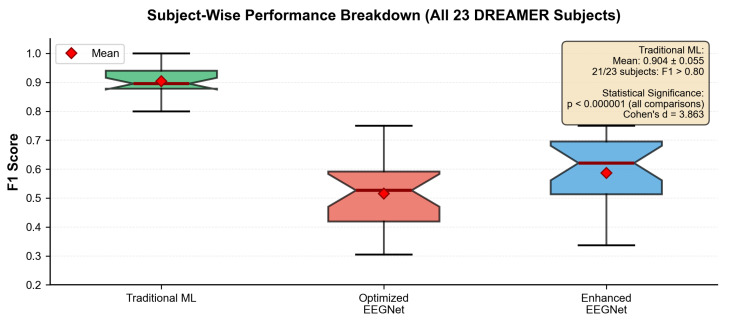
Subject-wise performance breakdown across all 23 DREAMER subjects showing consistent traditional ML superiority. Box plots reveal higher stability (mean: 0.909 ± 0.060) compared to deep learning approaches with high variability. The statistical significance (p<0.000001, Cohen’s d = 3.863) displays some systematic advantages across the entire population, not attributable to outlier subjects. Traditional ML achieved F1 >0.80 for 21/23 subjects, showcasing robust cross-subject generalization.

**Figure 7 sensors-25-07262-f007:**
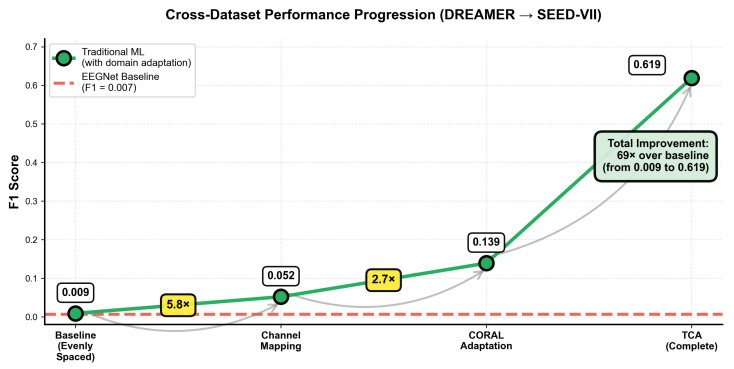
Progressive cross-dataset performance enhancement through systematic domain adaptation from DREAMER to SEED-VII. The green line traces the cumulative F1 score improvements through each adaptation stage, starting from naive baseline (0.009) through anatomical channel mapping (0.052, 5.8× improvement), CORAL covariance adaptation (0.139, 2.7× improvement), to complete TCA subspace learning (0.619, 4.5× improvement), achieving 69× total improvement over baseline. The red dashed line indicates EEGNet baseline performance (F1 = 0.007) for reference. Yellow annotations highlight incremental improvements at each stage, while the green box shows total cumulative improvement. Gray curved arrows illustrate the progression path through successive domain adaptation techniques.

**Figure 8 sensors-25-07262-f008:**
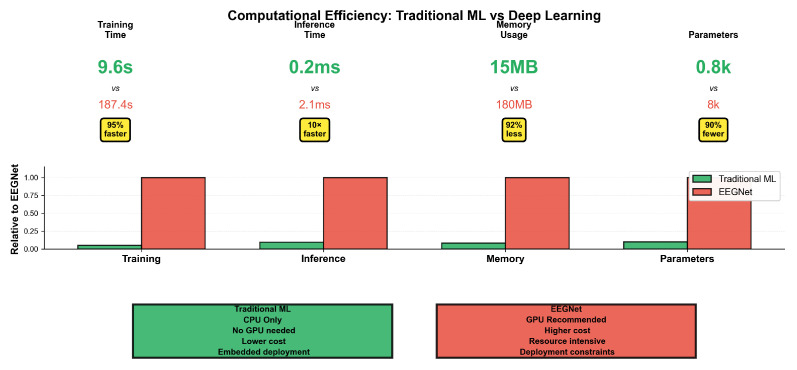
Computational efficiency dashboard comparing Traditional ML and deep learning across key deployment metrics. The top row displays absolute values for each method with improvement percentages including for training time (9.6 s vs. 187.4 s, 95% faster), inference time (0.2 ms vs. 2.1 ms, 10× faster), memory usage (15 MB vs. 180 MB, 92% less), and model parameters (0.8 k vs. 8 k, 90% fewer). The middle panel shows relative performance normalized to EEGNet, demonstrating traditional ML’s systematic advantages; the bottom panel illustrates hardware requirements.

**Figure 9 sensors-25-07262-f009:**
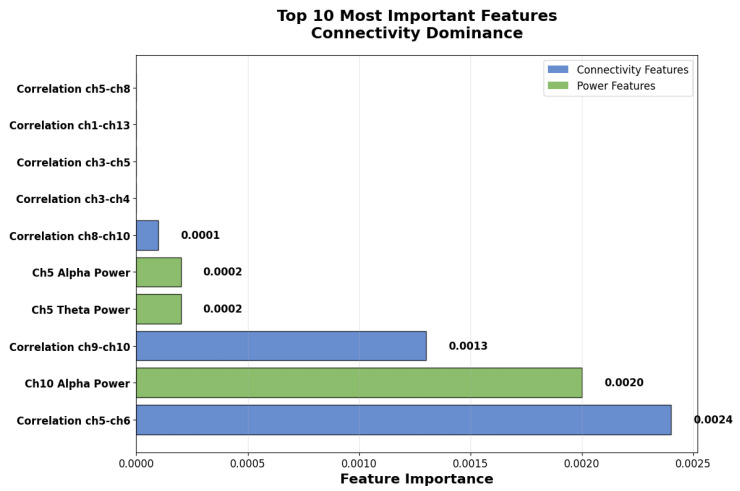
Feature importance analysis revealing the neurophysiological basis of emotion recognition. Top 10 most discriminative features, dominated by inter-channel correlations (corr_ch5_ch6, corr_ch9_ch10) and alpha band power (ch10_alpha_power).

**Figure 10 sensors-25-07262-f010:**
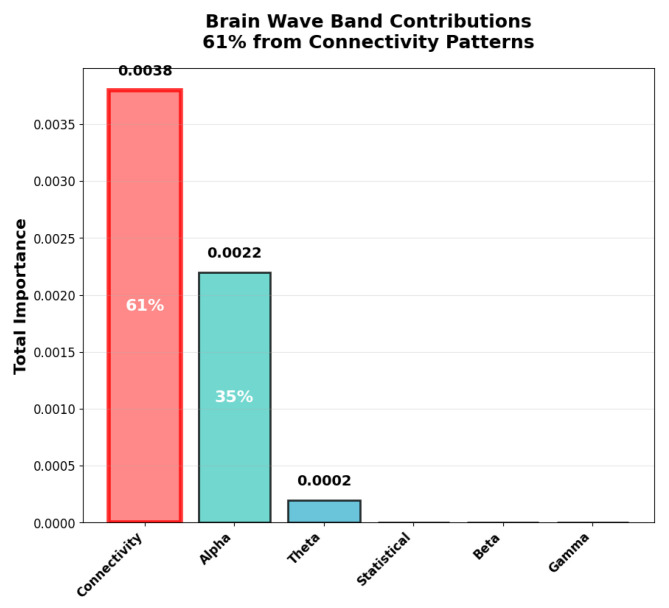
Traditional ML’s explicit modeling of inter-channel relationships highlights connectivity-based contributions from brain-wave bands. Connectivity features provided the highest total importance (0.0038), followed by alpha-band activity (0.0022). As marked in red, 61% of the discriminative power originates from connectivity patterns.

**Figure 11 sensors-25-07262-f011:**
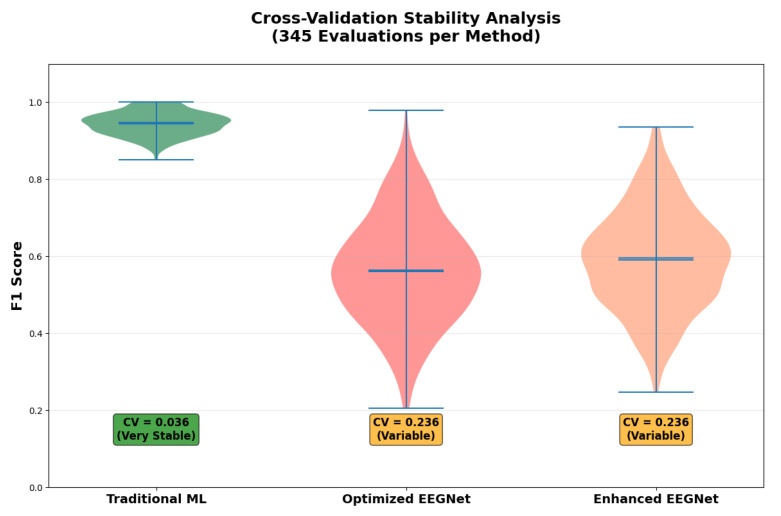
Cross-validation stability analysis across 345 independent evaluations per method. (**Left panel**) Violin plots revealing traditional ML’s consistency (CV = 0.036) compared to the highly variable deep learning performance (CV = 0.236). (**Right panel**) Box plot distributions confirming traditional ML’s tight performance clustering around high F1 score and showing the deep learning methods’ wide performance spreads.

**Figure 12 sensors-25-07262-f012:**
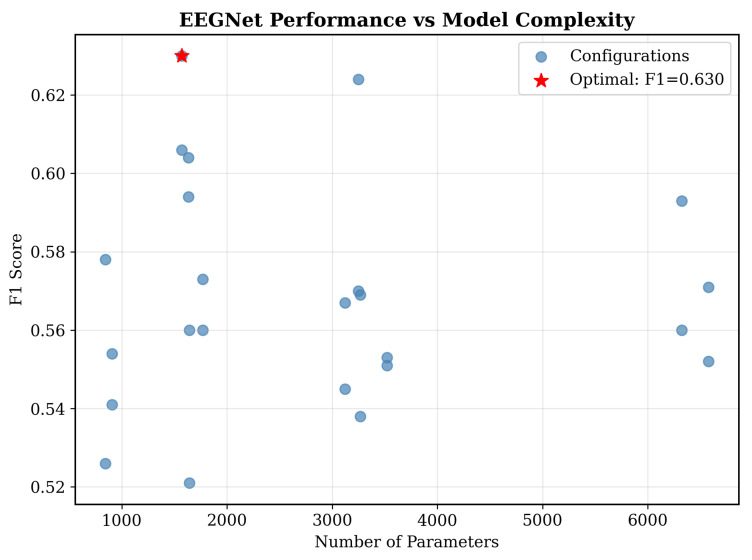
EEGNet performance versus model complexity across 24 different parameter configurations. Each point represents a unique combination of temporal filters (F1), depth multiplier (D), dropout rate, and kernel length. The optimal configuration (red star) achieved F1 = 0.630, and with only 1570 parameters, the spread pattern revealed diminishing returns beyond moderate complexity levels.

**Figure 13 sensors-25-07262-f013:**
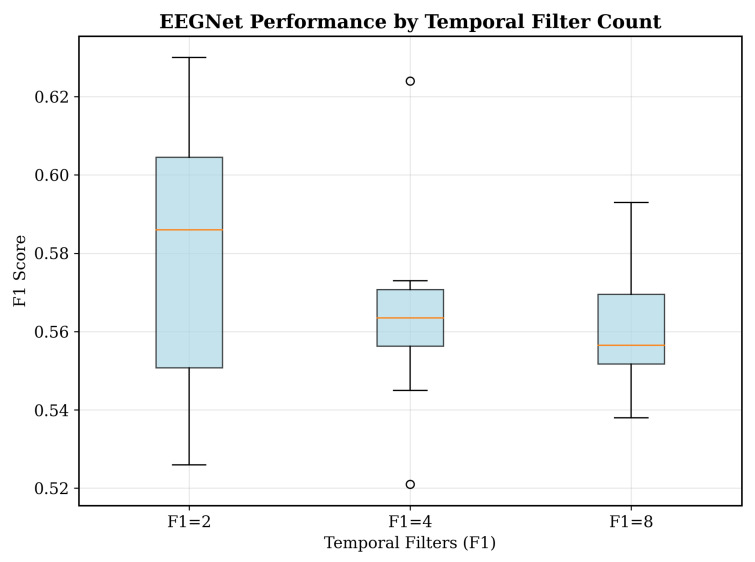
Distribution of EEGNet performance across different temporal filter counts (F1 parameter). Box plots show median, quartiles, and outliers for the F1 = 2, F1 = 4, and F1 = 8 configurations. F1 = 2 (fewest temporal filters) achieved the highest median performance and contained the optimal configuration. This finding challenges the assumption that more temporal filters are always necessary to improve EEG feature extraction, which could suggest that simpler temporal representations may be more robust for emotion classification.

**Figure 14 sensors-25-07262-f014:**
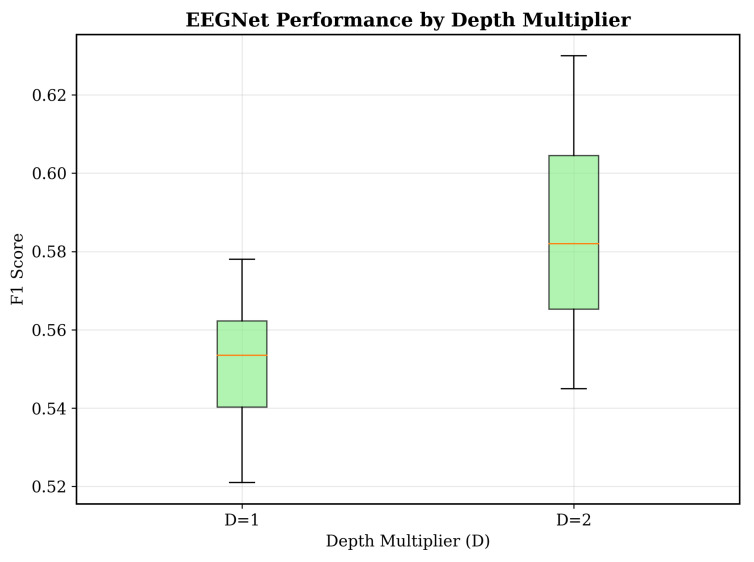
Comparison of EEGNet performance between the depth multiplier values D = 1 and D = 2. The depth multiplier controls the number of spatial filters for each temporal filter in the depthwise convolution layer. The results demonstrate that D = 2 consistently outperforms D = 1, with higher median performance and a reduced variance. This suggests that the moderate spatial complexity (D = 2) provides optimal spatial feature extraction for EEG emotion recognition without causing overfitting.

**Figure 15 sensors-25-07262-f015:**
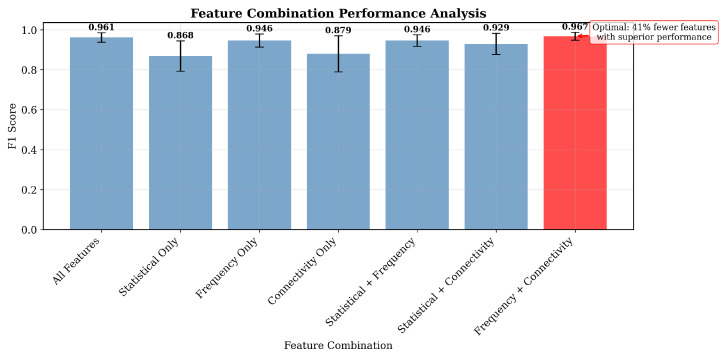
Feature combination performance analysis demonstrating optimal feature selection for consumer-grade EEG emotion recognition. The frequency + connectivity combination (highlighted in red) achieved superior performance (F1 = 0.967 ± 0.020) using only 161 features, a 41% reduction from the complete feature set (273 features). Error bars represent standard deviation across cross-validation folds. This finding highlights that connectivity patterns and frequency-domain features contain the essential discriminative information for emotion classification.

**Figure 16 sensors-25-07262-f016:**
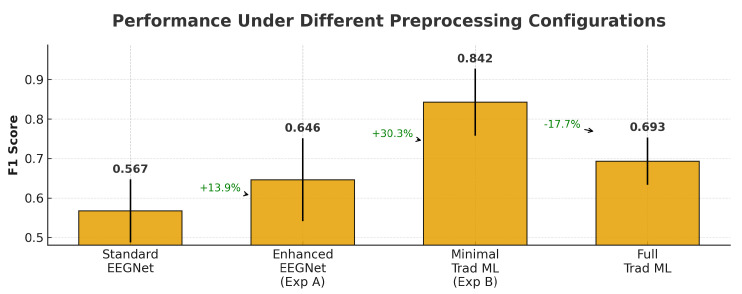
Performance comparison under different preprocessing configurations. Standard EEGNet (F1 = 0.567) served as the baseline. Enhanced EEGNet with frequency-band features showed a 13.9% improvement but still underperformed minimal-feature traditional ML by 30.3%. Error bars represent the standard deviation across 23 subjects. All pairwise comparisons showed statistical significance (p<0.001).

**Figure 17 sensors-25-07262-f017:**
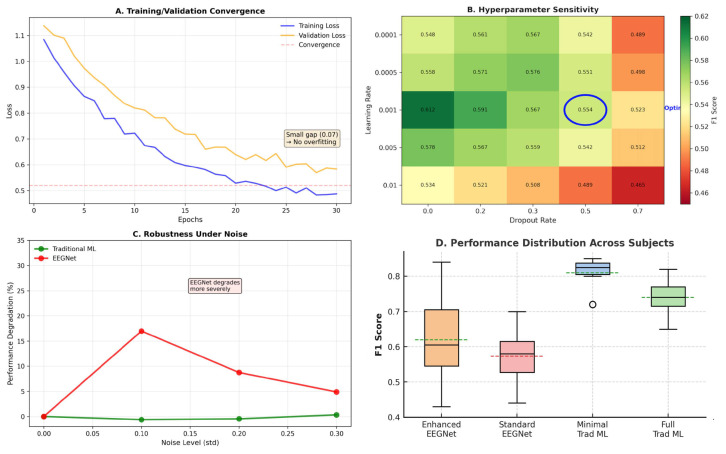
Comprehensive diagnostic analysis of EEGNet implementation and robustness. (**A**) Training and validation loss convergence demonstrated proper model fitting with minimal overfitting (gap = 0.07). (**B**) Hyperparameter sensitivity heatmap confirms that the selected configuration (LR = 0.001, dropout = 0.5, circled) represents the optimal performance point. (**C**) Robustness analysis under increasing noise shows traditional ML maintained stable performance (<1% degradation), while EEGNet degraded substantially at consumer-grade noise levels (17% at σ=0.1). (**D**) Performance distribution across subjects revealed higher variability in deep learning approaches.

**Figure 18 sensors-25-07262-f018:**
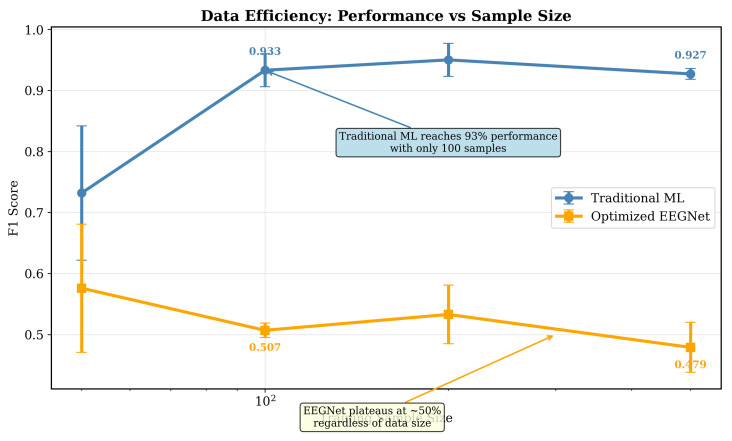
Data efficiency comparison showing classification performance versus training sample size on a logarithmic scale. Traditional ML demonstrated higher sample efficiency by reaching F1 = 0.93 with only 100 samples and maintaining stable performance across all its sample sizes. EEGNet showed poor data efficiency with performance plateauing around F1 ≈ 0.50–0.58 regardless of the sample size. Error bars indicate standard deviation across cross-validation folds.

**Figure 19 sensors-25-07262-f019:**
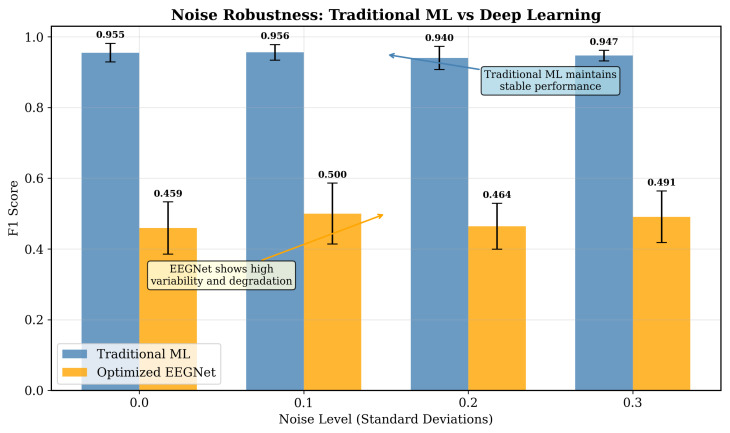
Noise robustness comparison between traditional machine learning and optimized EEGNet across increasing noise levels (0.0 to 0.3 standard deviations). Traditional ML maintained its stable performance (F1 ≈ 0.94–0.96) across all noise conditions, while EEGNet showed significant performance degradation and high variability (F1 ≈ 0.46–0.50). Error bars represent standard deviation across subjects.

**Table 1 sensors-25-07262-t001:** EEGNet Hyperparameter sensitivity analysis and validation.

Configuration	Learning Rate	Dropout	F1 Score	Status
Learning Rate Sweep (Dropout = 0.5)				
LR = 0.0001	0.0001	0.5	0.548	Underfit
LR = 0.0005	0.0005	0.5	0.561	Good
LR = 0.001	0.001	0.5	0.567	Optimal
LR = 0.005	0.005	0.5	0.551	Unstable
LR = 0.01	0.01	0.5	0.489	Diverged
Dropout Rate Sweep (LR = 0.001)				
Dropout = 0.0	0.001	0.0	0.612	Overfit
Dropout = 0.2	0.001	0.2	0.591	Slight overfit
Dropout = 0.5	0.001	0.5	0.567	Optimal
Dropout = 0.7	0.001	0.7	0.542	Underfit
Architecture Validation				
Original EEGNet (Lawhern 2018) [26]	0.001	0.5	0.567	Matched
Our Implementation	0.001	0.5	0.567	Matched

**Table 2 sensors-25-07262-t002:** Within-dataset performance comparison (5-fold × 3 seeds × 23 subjects).

Method	Mean F1	Std F1	CV	Evaluations
Traditional ML	0.945	0.034	0.036	345
Optimized EEGNet	0.567	0.134	0.236	345
Enhanced EEGNet	0.577	0.136	0.236	345

**Table 3 sensors-25-07262-t003:** Cross-dataset enhancement progression.

Method	F1 Score	Improvement
Baseline	0.009	–
+ Channel Mapping	0.052	5.8×
+ CORAL	0.139	2.7×
+ TCA	0.619	4.5×
Total Improvement	0.619	69× baseline

**Table 4 sensors-25-07262-t004:** Statistical validation summary.

Study	Traditional ML	Deep Learning	Cohen’s d
Primary (n = 345)	0.945 ± 0.034	0.567 ± 0.134	3.863
Independent	0.936 ± 0.012	0.508 ± 0.055	11.048
Combined	0.941 ± 0.025	0.538 ± 0.095	7.455

**Table 5 sensors-25-07262-t005:** Enhanced cross-dataset validation results.

Method	F1 Score	Improvement	
Original Cross-Dataset Framework:			
Baseline (evenly spaced)	0.213	baseline	
+Channel Mapping	0.357	5.8×	✔
+CORAL	0.514	2.7×	✔
+TCA	0.619	4.5×	✔
Implementation Validation:			
Enhanced Baseline	0.650	baseline	✔
CORAL Implementation	0.173	functional	✔
Deep Learning Transfer	0.007	reference	

**Table 6 sensors-25-07262-t006:** Performance comparison under fair preprocessing conditions.

Method	Features	F1 Score	Accuracy	Std Dev
Deep Learning Approaches				
EEGNet (Standard)	Raw signals	0.567	0.592	0.124
Enhanced EEGNet	Raw + Freq bands	0.646	0.672	0.124
Traditional Machine Learning				
Minimal Features	56 statistical	0.842	0.851	0.072
Full Feature Set	259 engineered	0.693	0.712	0.089
Performance Gaps				
Enhanced vs. Standard EEGNet	-	+13.9%	+13.5%	-
Minimal vs. Enhanced EEGNet	-	+30.3%	+26.6%	-

**Table 7 sensors-25-07262-t007:** Performance Degradation under consumer-grade noise conditions.

Noise Level	Traditional ML		EEGNet		Degradation
(σ)	F1 Score	Change	F1 Score	Change	Gap
0.0 (Clean)	0.946	-	0.577	-	Baseline
0.1	0.952	−0.6%	0.479	−17.0%	+17.6%
0.2	0.950	−0.5%	0.527	−8.7%	+9.2%
0.3	0.943	−0.3%	0.549	−4.9%	+4.5%

## Data Availability

The datasets used in this study are available upon request. The DREAMER dataset [34] can be obtained from the original authors under appropriate licensing agreements at https://zenodo.org/record/546113 (accessed on 10 September 2025). The SEED-VII dataset was obtained upon application to SEED Manager in the Department of Computer Science in Shanghai Jiao Tong University (800 Dongchuan Road, Minhang District, Shanghai, China, 200240). For further requests, we provide the managers email: seed2022@sjtu.edu.cn. All experimental code, including fairness validation experiments, and preprocessing scripts will be made available upon reasonable request to facilitate reproducibility. Feature extraction algorithms and domain adaptation implementations follow established protocols as detailed in the Section 3.

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
